# Universal Sharpness Dynamics in Neural Network Training: Fixed Point Analysis, Edge of Stability, and Route to Chaos

**Dayal Singh Kalra** [2]**, Tianyu He** [1] **& Maissam Barkeshli** [1, 3]
{dayal,tianyuh,maissam}@umd.edu

## Abstract

In gradient descent dynamics of neural networks, the top eigenvalue of the loss Hessian (sharpness) displays a variety of robust phenomena throughout training. This includes early time regimes where the sharpness may decrease during early periods of training (sharpness reduction), and later time behavior such as progressive sharpening and edge of stability. We demonstrate that a simple 2-layer linear network (UV model) trained on a single training example exhibits all of the essential sharpness phenomenology observed in real-world scenarios. By analyzing the structure of dynamical fixed points in function space and the vector field of function updates, we uncover the underlying mechanisms behind these sharpness trends. Our analysis reveals (i) the mechanism behind early sharpness reduction and progressive sharpening, (ii) the required conditions for edge of stability, (iii) the crucial role of initialization and parameterization, and (iv) a period-doubling route to chaos on the edge of stability manifold as learning rate is increased. Finally, we demonstrate that various predictions from this simplified model generalize to real-world scenarios and discuss its limitations.

## 1 Introduction

Over the last several years, it has been observed that the training dynamics of neural networks exhibits a rich and robust set of unexpected phenomena, stemming from the non-convexity of the loss landscape. These phenomena not only challenge our existing understanding of loss landscapes but also open avenues for significantly enhancing model performance through improved optimization techniques. In particular, the unexpected and robust phenomenology is mainly associated with the evolution of the Hessian of the loss function, which provides a measure of the local curvature of the loss landscape and plays an important role in understanding generalization performance Keskar et al. (2016); Dziugaite & Roy (2017); Jiang et al. (2019). However, the relationship between sharpness and generalization has been called into question Dinh et al. (2017); Kaur et al. (2023).

On the one hand, it has been observed that at late training times, gradient descent (GD) typically exhibits "progressive sharpening," where the top eigenvalue of the loss Hessian $\lambda^H$, referred to as the sharpness, gradually increases with time, until it reaches roughly $2/\eta$, where $\eta$ is the learning rate. Once the sharpness reaches roughly $2/\eta$, it stops increasing and typically oscillates near $2/\eta$, a late-time training phenomenon referred to as the "edge of stability (EoS)" Cohen et al. (2021). On the other hand, during early training, a decrease in sharpness is observed —referred to as "sharpness reduction" Kalra & Barkeshli (2023) —before hitting a temporary plateau.

For large enough learning rates, training temporarily destabilizes early on, and the network "catapults" out of its local basin, leading to a temporary sudden increase in the loss in the first few steps, before eventually settling down in a flatter region of the loss landscape characterized by lower sharpness Lewkowycz et al. (2020). Similar to the loss, sharpness may also spike within the first few steps of training and quickly decrease (sharpness catapult). A rich phase diagram as a function of network depth, width and learning rate summarizes the early training dynamics Kalra & Barkeshli (2023).

[1]Department of Physics, University of Maryland, College Park
[2]Department of Computer Science, University of Maryland, College Park
[3]Joint Quantum Institute, University of Maryland, College Park

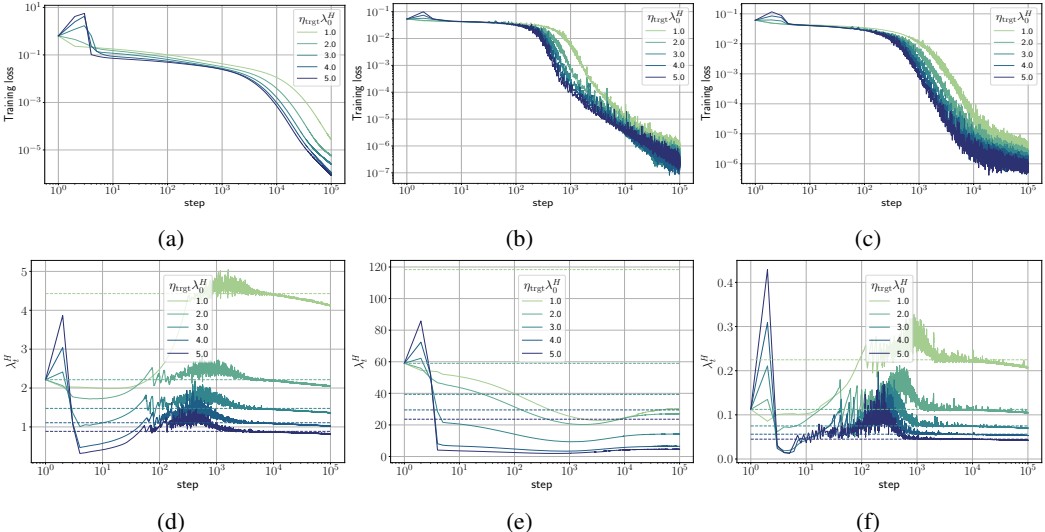

Figure 1: Training loss and sharpness trajectories of ReLU FCNs trained on a 5k subset of CIFAR-10 examples using MSE loss and GD: (a, d) SP with $\sigma_w^2 = 0.5$, (b, e) SP with $\sigma_w^2 = 2.0$, (c, f) $\mu$P with $\sigma_w^2 = 2.0$. The dashed lines in the sharpness figures show the $2/\eta$ threshold.

The discovery of these intriguing sharpness phenomena has attracted significant attention, with an emphasis on various toy models that exhibit similar phenomenology. Yet, the specific conditions and reasons why these phenomena occur still remain elusive. In this paper, we analyze a simple toy model, a 2-layer linear network trained on one example, referred to as the UV model. We show that all of the phenomena described above can be observed in the UV model for appropriate choices of learning rate, initialization, parameterization, and choice of training example. Through this exploration, we provide novel insights into the mechanisms at play and offer predictions that we validate in realistic architectures with both real and synthetic datasets.

**Our Contributions.** We revisit the four training regimes identified by Kalra & Barkeshli (2023) (early time transient, intermediate saturation, progressive sharpening, and late time EoS) in Section 3, focusing on the crucial role of initializations and parameterizations. Our findings reveal that models in Standard Parameterization (SP) with large initializations do not exhibit EoS, even at late training times. Moreover, we show that models in Maximal Update Parameterization ($\mu$P) Yang & Hu (2021) do not experience an early sharpness reduction. This result also holds for models in SP with small initializations.

We show the UV model exhibits all four training regimes and also captures the effect of initializations and parameterization discussed above. Through fixed-point analysis of the UV model in the function space, we analyze the origins of the various dynamical phenomena exhibited by the sharpness. Specifically, we demonstrate in Sections 4 and 5: (i) the emergence of various sharpness phenomena arising from the stability and position of the dynamical fixed points, (ii) a critical learning rate $\eta_c$, above which the model exhibits EoS on a sub-quadratic manifold, and (iii) a period-doubling route to chaos of sharpness fluctuations as learning rate is increased in the EoS regime.

In Section 6, we verify various non-trivial predictions from the UV model in realistic architectures with real and synthetic datasets. Our findings reveal: (i) a sharpness-weight norm correlation before the training enters the EoS regime, (ii) a phase diagram of EoS, revealing initializations and parameterizations that do not exhibit EoS, and (iii) a period-doubling route to chaos in real architectures trained on synthetic datasets, while those trained on real datasets exhibit long-range correlations at the EoS, with a remnant of the period doubling route to chaos.

**Related Works.** Using the top eigenvalue of the Neural Tangent Kernel (NTK) $\lambda^K$ at initialization ($t = 0$), Lewkowycz et al. (2020) revealed a 'catapult phase', $2/\lambda_0^K < \eta < \eta_{\max}$, in which training converges despite an initial spike in training loss. After the early training phase, sharpness continues to increase until it reaches a break-even point (Jastrzebski et al., 2020), beyond which GD dynamics typically enters the EoS regime (Cohen et al., 2021). This has motivated various theoretical studies

to understand GD dynamics at large learning rates Ma et al. (2022); Wang et al. (2022); Arora et al. (2022); Damian et al. (2023); Rosca et al. (2023); Zhu et al. (2023); Wu et al. (2023); Chen & Bruna (2023); Ahn et al. (2022); Kreisler et al. (2023); Song & Yun (2023); Chen et al. (2023). These include an analysis by Wang et al. (2022), who show analyzed EoS in a 2-layer linear network using the norm of the last layer, restricting to cases that show progressive sharpening from initialization. Agarwala et al. (2022) suggested that the UV model does not exhibit EoS behavior but showed that a modified model exhibits progressive sharpening and two-step oscillations at EoS using NTK as the proxy. In contrast, we show that the UV model shows EoS behavior under the appropriate choice of parameterization and training example. Meanwhile Chen & Bruna (2023) analyzed two-step gradient updates of a single-neuron network and matrix factorization to gain insights into EoS. Furthermore, Song & Yun (2023) analyzed a 2-layer linear network under logistic loss and demonstrated that sharpness at late training times oscillates around $2f/\eta\ell'$, where $f$ is the network output and $\ell'$ is the derivative of the loss. Another study by Chen et al. (2023) analyzed large learning rate dynamics of toy models which are characterized by a one-dimensional cubic map. Our work, in contrast, delves into various sharpness phenomena occurring throughout the training trajectory and analyzes their origins. It is worth noting that a concurrent study by Wang et al. (2023) also examines sharpness throughout training. Noci et al. (2024) relate sharpness dynamics to learning rate transfer in $\mu$P networks by showing that sharpness trajectories do not change appreciably when depth and width are varied. Given that our analysis spans the entire training trajectory, it relates to numerous studies. Hence, we defer a comprehensive discussion of related works to Appendix A.

## 2 NOTATIONS AND PRELIMINARIES

This section describes the fundamental concepts and notations that form the basis of our analysis.

**Dynamical Systems and Fixed Points:** Consider a discrete dynamical system described by $\boldsymbol{\theta}_{t+1} = M(\boldsymbol{\theta}_t)$. A fixed point $\boldsymbol{\theta}^*$ of the dynamics satisfies $M(\boldsymbol{\theta}^*) = \boldsymbol{\theta}^*$. The linear stability of a fixed point $\boldsymbol{\theta}^*$ is determined by analyzing the eigenvalues $\{\lambda_i^{J^*}\}$ of the Jacobian $J_M(\boldsymbol{\theta}^*) \coloneqq \nabla_\theta M(\theta)\,|_{\theta=\theta^*}$. An eigendirection $u_i^{J^*}$ of a fixed point $\theta^*$ is stable if $|\lambda_i^{J^*}| < 1$ and unstable if $|\lambda_i^{J^*}| > 1$ Ott (2002). The dynamics is captured by the vector field of updates $G(\theta) \coloneqq M(\theta) - \theta$. The corresponding unit vector is denoted $\hat{G}(\theta) \coloneqq {}^{G(\theta)}\!/\!{}_{\|G(\boldsymbol{\theta})\|}$. Nullclines refer to curves where one of the variables, $\theta_i$, remains invariant, i.e., $\theta_{i;t} = M_i(\boldsymbol{\theta_t})$.

**Parameterizations in Neural Networks:** Sharpness phenomena in neural networks are intrinsically tied to network parameterization. Standard Parameterization (SP) Sohl-Dickstein et al. (2020) and Neural Tangent Parameterization (NTP) Jacot et al. (2018) are two commonly used parameterizations, which converge to kernel methods at infinite width. Yang & Hu (2021) proposed Maximal update Parameterization ($\mu$P), which allows for feature learning at infinite width. For implementation details, see Appendix B.2.1.

**UV Model:** The UV model refers to a 2-layer linear network $f : \mathbb{R}^d \to \mathbb{R}$ trained on a single example. We parameterize $f$ as $f(\boldsymbol{x};\theta) = \frac{1}{\sqrt{n^{1-p}}}\boldsymbol{v}^T U \boldsymbol{x}$, where $\boldsymbol{x} \in \mathbb{R}^d$ is the input, $n$ is the network width, and $\boldsymbol{v} \in \mathbb{R}^n, U \in \mathbb{R}^{n \times d}$ are trainable parameters, with each component drawn i.i.d. at initialization from a normal distribution $\mathcal{N}(0, \sigma_w^2/n^p)$. Here, $p \in [0,1]$ is a parameter that interpolates between NTP and $\mu$P, and $n_{\text{eff}} \coloneqq n^{1-p}$ is referred to as the effective width. We consider the network trained on a single training example $(\boldsymbol{x}, y)$ using MSE loss $\ell\left(f(\boldsymbol{x};\theta), y\right) = \frac{1}{2}\left(f(\boldsymbol{x};\theta) - y\right)^2$.

## 3 REVIEW OF THE FOUR REGIMES OF TRAINING

Typical training trajectories of neural networks can be categorized into four training regimes Kalra & Barkeshli (2023), as shown in Figure 1(a, d):

(T1) *Early time transient:* This corresponds to the first few steps of training. At small learning rates ($\eta < \eta_{\text{loss}}$), loss and sharpness decrease monotonically. At larger learning rates ($\eta > \eta_{\text{loss}}$), training catapults out of the initial basin, temporarily increasing the loss, and finally converges to a flatter region Lewkowycz et al. (2020). By the end of this regime, sharpness has decreased from initialization for all learning rates, and more substantially at larger learning rates.

(T2) *Intermediate saturation:* Following the initial transient regime, sharpness approximately plateaus before gradually increasing.

(T3) *Progressive sharpening:* In this regime, sharpness continues to increase until it reaches $\lambda^H \approx 2/\eta$ Jastrzebski et al. (2020); Cohen et al. (2021). At large effective widths or small learning rates, training may conclude before reaching this threshold.

(T4) *Late-time dynamics (EoS):* After progressive sharpening, for MSE loss, sharpness oscillates around $2/\eta$. For cross-entropy loss, the sharpness oscillates when reaching approximately $2/\eta$, while decreasing over longer time scales Cohen et al. (2021).

In this work, we show that the sharpness dynamics heavily depends on the initialization and parameterization of the network, and not every training trajectory shows all four regimes. For instance, Figure 1(b, e) shows that FCNs in SP with large initialization (or large effective width) do not exhibit EoS, even when loss decreases to a value below $10^{-5}$. Following the early transient regime, sharpness monotonically decreases, with only a nominal increase towards late training. In contrast, Figure 1(c, f) shows that FCNs in $\mu$P (or small effective width) do not experience an initial sharpness reduction at small learning rates ($\eta < \eta_{\text{loss}}$). Rather, sharpness continues to increase until it reaches $2/\eta$ and then oscillates around it. At large learning rates ($\eta > \eta_{\text{sharp}}$), sharpness catapults and eventually settles into the same trend as above.

These different training regimes are generically observed for more complex architectures and datasets as we show in Appendix D.3, including CNNs and ResNets, trained on CIFAR-10 and Transformers trained on Wikitext-2.

In Appendix D.1, we show that these trends remain consistent when NTP is used instead of SP. Given this similarity in the training dynamics between SP and NTP, we use NTP for theoretical analysis for clarity and SP in realistic experiments for implementation convenience.

Figure 2 (and Figure 7 in Appendix C.5) demonstrates that the UV model displays all four training regimes. It also captures the cases where sharpness reduction or EoS is not observed. Therefore, the simplified UV model can serve as an effective model for understanding these universal behaviors in the sharpness dynamics. In the subsequent section, we perform fixed point analysis of the UV model and probe the origin of these complex phenomena in later sections.

## 4 FIXED POINT ANALYSIS OF THE UV MODEL

Under GD, the parameters of the UV model are updated as $U_{t+1} = U_t - \eta \frac{\Delta f_t \boldsymbol{v}_t \boldsymbol{x}^T}{\sqrt{n_{\text{eff}}}}$ , $\boldsymbol{v}_{t+1} = \boldsymbol{v}_t - \eta \frac{\Delta f_t U_t \boldsymbol{x}}{\sqrt{n_{\text{eff}}}}$, where $\eta$ is the learning rate and $\Delta f_t := f(\boldsymbol{x}; \theta_t) - y$ is the residual at training step $t$. In function space, the dynamics can be completely described using the residual $\Delta f_t$ and trace of the loss Hessian $\lambda := \text{Tr } H = \frac{1}{n_{\text{eff}}} \left( \boldsymbol{x}^T U^T U \boldsymbol{x} + \boldsymbol{v}^T \boldsymbol{v} \, \boldsymbol{x}^T \boldsymbol{x} \right)$, which is also the scalar neural tangent kernel in this case. The function space dynamics of the UV model can be fully described using two coupled non-linear equations (for derivation, see Appendix C.1):

$$\Delta f_{t+1} = \Delta f_t \left( 1 - \eta \lambda_t + \frac{\eta^2 \|\boldsymbol{x}\|^2}{n_{\text{eff}}} \Delta f_t (\Delta f_t + y) \right), \tag{1}$$

$$\lambda_{t+1} = \lambda_t + \frac{\eta \|\boldsymbol{x}\|^2}{n_{\text{eff}}} \Delta f_t^2 \left( \eta \lambda_t - 4 \frac{(\Delta f_t + y)}{\Delta f_t} \right), \tag{2}$$

with effectively three parameters: $\eta$, $\|\boldsymbol{x}\|/\sqrt{n_{\text{eff}}}$ and $y$. While similar equations have been considered in previous works Lewkowycz et al. (2020); Zhu et al. (2022); Agarwala et al. (2022), the generalization to generic parameterizations is novel and would be crucial in observing different sharpness phenomena such as EoS. The $y = 0$ case has been analyzed in prior works Lewkowycz et al. (2020); Kalra & Barkeshli (2023) for understanding catapult dynamics. Here, $\lambda$ can only decrease with time, as can be seen from Equation (2) with $\eta < \eta_{\max} = 4/\lambda_0$ (training diverges if $\eta > \eta_{\max}$). As a result, the model does not exhibit progressive sharpening and EoS. Below we focus on the case $y > 0$, which allows for $\lambda$ to increase in time and consequently, much richer dynamics.

Equations (1) and (2) have four distinct fixed points/lines (referred to as I-IV) as detailed in Table 1 of Appendix C.3. The fixed line I defines a zero-loss line, meaning $\ell = 0$ for all points in I; the points

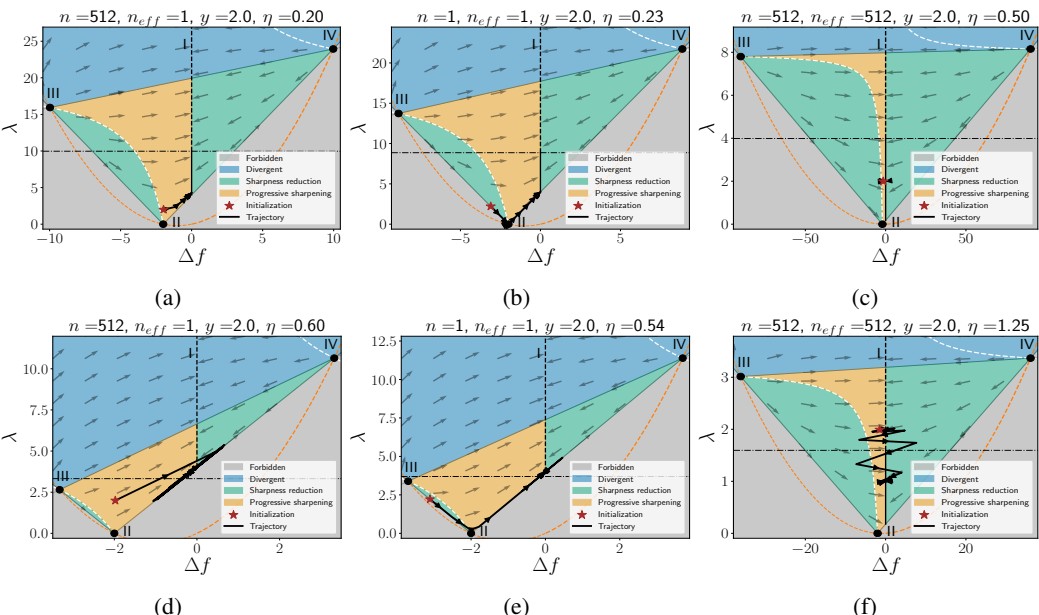

Figure 2: Training trajectories of the UV model with $\|x\| = 1$ and $y = 2$ in the $(\Delta f, \lambda)$ plane for different values of $n$, $n_{\text{eff}}$ and $\eta$. The columns show initializations with different $n$ and $n_{\text{eff}}$, while the rows represent increasing learning rates for fixed initializations. The horizontal dash-dot line $\eta\lambda = 2$ separates the stable (solid black vertical line) and unstable (dashed black vertical line) fixed points along the zero loss fixed line I. Forbidden regions, $2\|x\|\|\Delta f + y\|/\sqrt{n_{\text{eff}}} > \lambda$, (see Appendix C.2) are shaded gray. The nullclines $\Delta f_{t+1} = \Delta f_t$ and $\lambda_{t+1} = \lambda_t$ are shown as orange and white dashed curves, respectively. Sharpness reduction, progressive sharpening, and divergent regions are colored green, yellow, and blue. The gray arrows indicate the local vector field $\hat{G}(\Delta f, \lambda)$, which is the direction of the updates. The training trajectories are depicted as black lines with arrows, with the star marking the initialization. In all cases, $\eta_c = \sqrt{n_{\text{eff}}}/2$ (introduced in Section 5.2).

in I are stable for $\eta\lambda < 2$ and unstable otherwise. Fixed point II at $(-y, 0)$ corresponds to the origin in parameter space ($U, \mathbf{v} = 0$) and it is a saddle point of the dynamics for convergent learning rates $\eta$. Both I and II are also fixed points of the GD optimization, i.e., critical points of the loss. The loss Hessian at I is positive definite, while fixed point II is a saddle point in the loss landscape. The remaining two fixed points III and IV are unstable and exist only in function space, representing non-trivial parameter space dynamics that leave the function space dynamics invariant.

Figure 2 shows the fixed points and the vector field $\hat{G}(\Delta f, \lambda)$ determined by Equations (1) and (2), which illustrates the direction of the updates at each point. Note that the stability of the fixed line (I) does not follow from $\hat{G}$ alone, as the magnitude $G$ is required to determine stability. Figure 2 also shows training trajectories for various parameter values. Using $\lambda$ as a proxy for sharpness, we see there are regions where $\lambda$ increases (colored yellow) and decreases (colored green) along the flow, which we refer to as progressive sharpening and sharpness reduction, respectively. It follows from Equation (2) that the condition $\eta\lambda\Delta f = 4(\Delta f + y)$ separates these regions. Importantly, the parameters $\eta$, $\|x\|/\sqrt{n_{\text{eff}}}$ and $y$ influence the position of the fixed points. This, in turn, affects the extent of different regions and the vector field $\hat{G}$, as illustrated in Figure 2. In particular, on decreasing effective width $n_{\text{eff}}$, or increasing learning rate $\eta$, fixed points III and IV move inward (see fixed point expressions in Table 1), which relatively enlarges the progressive sharpening region while shrinking the overall convergent region. Overall, these illustrations demonstrate how the local stability and relative position of the fixed points collectively impact the dynamics. In the subsequent section, we discuss the dynamics in detail.

## 5 Understanding Sharpness Dynamics in the UV model

In this section, we describe the origin of different robust phenomena in the dynamics of sharpness using the fixed point and linear stability analysis from the previous section. This explains the four training regimes observed in the UV model. We will discuss the influence of effective width and initializations, shedding light on the differences between NTP and $\mu$P. For simplicity, we assume $\|\boldsymbol{x}\| = 1$, while allowing $n_{\text{eff}}$ to vary continuously. Note that we use $\lambda := \operatorname{Tr} H$ from the previous section as a proxy for sharpness; we have verified that the top eigenvalue of the Hessian of the loss also follows $\lambda$ (see Appendix C.5), although it is more difficult to analyze analytically.

### 5.1 Understanding Sharpness Trends Throughout Training

Figure 2 shows that the training dynamics can exhibit different behavior depending on the initial region. Below we summarize these based on empirical observations.

(R1) *Progressive sharpening region*: As shown in Figure 2(a, d), initialization in this region experiences an upward push due to the flow originating from fixed point II, resulting in a steady increase in $\lambda$. Depending on $\eta$ relative to a critical learning rate $\eta_c$ (introduced in Section 5.2) different late-time dynamics arises. For $\eta < \eta_c$, training converges to stable fixed points on the zero-loss line (I), as shown in Figure 2(a). When $\eta > \eta_c$, all points along the zero-loss line (I) become unstable, as shown in Figure 2(d). In this case, the network eventually converges to a line segment joining fixed points II and IV (the EoS manifold), where it continues to oscillate indefinitely between these fixed points, leading to the EoS phenomena. This will be analyzed in more depth in the subsequent section.

(R2) *Sharpness reduction region between fixed points II and III*: Figure 2(b, e) show that initializations in this region undergo a decrease in $\lambda$ as the flow is towards saddle point II. On approaching this saddle point, the dynamics slows down, resulting in the intermediate saturation regime. Eventually, training moves away from this saddle and enters the progressive sharpening region. From here on, the dynamics becomes akin to the case (R1).

(R3) *Sharpness reduction region b/w fixed line I and point IV*: Initializations in this region either converge to the nearby zero-loss solution for ($\eta < \eta_c$) or enter the progressive sharpening region for ($\eta > \eta_c$). In the latter case, the dynamics resembles those of case (R1).

So far, we have described the resultant dynamics when training is initialized in different regimes. Below, we describe the conditions that typically exhibit these regimes.

**Neural Tangent Parameterization:** In NTP, $\Delta f$ and $\lambda$ follow normal distributions: $\Delta f_0 \sim \mathcal{N}(-y, \sigma_w^4)$ and $\lambda_0 \sim \mathcal{N}(2\sigma_w^2, 4\sigma_w^4/n)$. Hence, the model can be initialized in any of the three regions described above. Moreover, fixed points III and IV move outward with increasing width, affecting the local vector field $\hat{G}(\Delta f, \lambda)$. At large widths $n \gg 1$, $\hat{G}(\Delta f_0, \lambda_0)$ at initialization points along $\begin{bmatrix} 1 & 0 \end{bmatrix}^T$ towards the zero-loss line. For small learning rates ($\eta < 2/\lambda_0$), training exponentially converges to the nearest zero-loss solution (see Figure 2(c)). Regardless of the initialization region, the change in $\lambda$ is minimal, receiving $\mathcal{O}(1/n)$ updates as per Equation (2). For large learning rates ($\eta > 2/\lambda_0$), the nearby zero-loss solution becomes unstable. Consequently, training catapults to a region with smaller $\lambda$, while bouncing between fixed points III and IV. This is the catapult effect studied in Lewkowycz et al. (2020) and Figure 2(f) demonstrates such a trajectory. By comparison, at small widths, the dynamics follows cases (R1-R3) discussed above.

**Maximal Update ($\mu$P) Parameterizations:** In contrast to NTP, the position of fixed points III-IV do not change with width $n$, and $\Delta f_0$ follows the distribution: $\Delta f_0 \sim \mathcal{N}(-y, \sigma_w^4/n)$, while $\lambda_0$ distribution remains unchanged. Consequently, at large widths, the model is initialized at $(-y, 2\sigma_w^2)$, right above fixed point II in the progressive sharpening region (R1), satisfying the condition $\eta \lambda_0 \Delta f_0 < 4(\Delta f_0 + y)$. Figure 2(a, d) shows such a trajectory. At small widths, fluctuations increase, making it plausible for $\mu$P networks to start in the sharpness reduction regions. In this case, the dynamics follow case (R2) or (R3).

### 5.2 Understanding Edge of Stability

This section analyzes the EoS behavior in the UV model, particularly from the fixed point perspective. As discussed in the previous section, the EoS behavior in the UV model arises when all fixed points

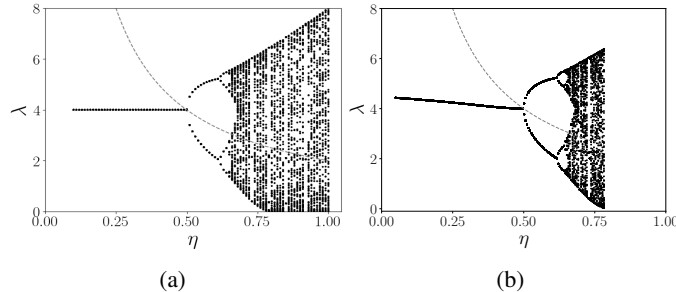

Figure 3: (a) Bifurcation diagram depicting limiting values of $\lambda$ obtained by simulating Equation (3). (b) Bifurcation diagram of the UV model. In both figures, $\|\boldsymbol{x}\| = 1$, $y = 2$, $n_{\text{eff}} = 1$ and $\eta_c = 0.5$.

along the zero-loss line (I) become unstable wrt the learning rate. Yet, the gradient updates (shown as gray arrows in Figure 2) continue to point towards the zero loss line. As a result, training is trapped in this region, converging to the line segment that joins fixed points II and IV —referred to as the EoS manifold —where it oscillates indefinitely.

**EoS Manifold is an Attractor:** By examining the two-step dyanmics akin to Agarwala et al. (2022); Chen & Bruna (2023), we show in Appendix C.7 that training converges to the EoS manifold above a critical learning rate $\eta_c$. For $\eta < \eta_c$, training converges to the stable fixed points on the zero-loss line. By comparison, for $\eta > \eta_c$, all points along the zero-loss line become unstable and the EoS manifold becomes a dynamical attractor. The critical $\eta_c$ for which all points on the zero-loss line become unstable thus gives a necessary condition for EoS:

**Result 1.** A necessary condition for the UV model to exhibit EoS is $\eta > \eta_c = \sqrt{n_{\text{eff}}}/\|\boldsymbol{x}\|y$ (see Appendix C.8 for details). It is useful to scale the learning rate as $\eta = c/\lambda_0$, in which case this condition becomes $\lambda_0 < c\|\boldsymbol{x}\|y/\sqrt{n_{\text{eff}}}$.

For learning rates $\eta > 2/\lambda_0$, training can catapult to regions with $\lambda_T < \lambda_0$. In such cases, the condition $\lambda_T < c\|\boldsymbol{x}\|y/\sqrt{n_{\text{eff}}}$ also applies.

**Dynamics on the EoS Manifold and Route to Chaos:** The dynamics on the EoS manifold satisfies $\lambda = 2\|\boldsymbol{x}\|(\Delta f + y)/\sqrt{n_{\text{eff}}}$, coupling $\Delta f$ and $\lambda$ together. This yields the map $\Delta f_{t+1} = M_f(\Delta f_t)$ describing the dynamics on the EoS manifold, with $M_f$ defined as

$$M_f(\Delta f_t) := \Delta f_t + \frac{\eta \Delta f_t}{\eta_c y}\left(\frac{\eta \Delta f_t}{\eta_c y} - 2\right)(\Delta f_t + y). \tag{3}$$

Figure 3(a) shows the limiting values of $\lambda$ (i.e. the values of $\lambda$ that the network jumps between at late times) as a function of learning rate, obtained by simulating Equation (3). We refer to this as the bifurcation diagram. As mentioned before, for $\eta > \eta_c$, the zero-loss solution becomes unstable with $\lambda$ oscillating around $2/\eta$ instead of converging. These fluctuations exhibit a fractal structure, as the system undergoes a series of period-doubling transitions with an increasing learning rate. This is the well-known *period-doubling route to chaos* Ott (2002). Figure 3(b) shows the bifurcation diagram of the UV model for $y = 2$. The bifurcation diagram diagram extends up to $\eta \approx 0.8$ before diverging at higher learning rates. This leads us to the following corollary of Result 1.

**Corollary 5.1.** *Let $\eta_{\max}$ be the maximum trainable learning rate for a given initialization. The bifurcation diagram is observed up to $\eta < \eta_{\max}$. If $\eta_{\max} < \eta_c$, the UV model does not exhibit EoS.*

These results suggest that models with small $\lambda_0$ and $n_{\text{eff}}$ are more prone to show EoS behavior. As a result, $\mu$P networks or those with small initial weight variance are more likely to exhibit EoS. On the other hand, large-width NTP networks may not show EoS behavior at all. In the next section, we will validate this prediction in real-world scenarios.

**Connections to sub-quadratic loss**: Ma et al. (2022) demonstrated that GD on sub-quadratic loss with large learning rates inherently results in EoS behavior. Here, we show that the loss on the EoS manifold of the UV model is sub-quadratic near its minimum. As noted above, the dynamics on the EoS manifold satisfies $\lambda = 2\|\boldsymbol{x}\|(\Delta f + y)/\sqrt{n_{\text{eff}}}$. The loss on the EoS manifold is then given by $\mathcal{L}(\theta) = \frac{1}{2}\Delta f^2 = \frac{y^2}{2}\left(\frac{\eta_c \lambda}{2} - 1\right)^2$ (see Appendix C.9 for derivation), where $\theta$ denotes the parameters.

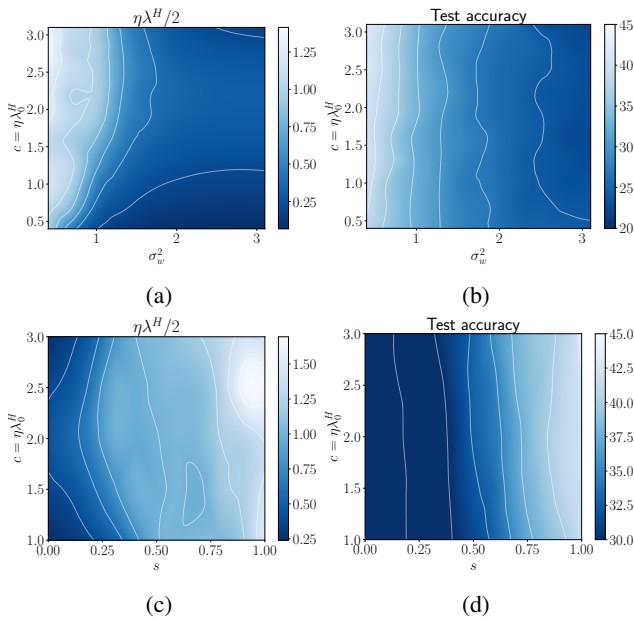

Figure 4: (a, b) Heatmap of $\eta\lambda^H/2$ and test accuracy of ReLU FCNs in SP trained on a 5k subset of CIFAR-10 until 99% training accuracy is achieved, with the weight variance $\sigma_w^2$ and learning rate multiplier $c = \eta\lambda_0^H$ as axes. As the color varies from blue to white, $\eta\lambda^H/2$ increases. (b, d) Same heatmaps with fixed $\sigma_w^2 = 2.0$, but varying $s$ continuously.

Since $\lambda \sim \mathcal{O}(\|\theta\|^2)$, the loss is of the form $\mathcal{L}(\theta) \approx \frac{1}{2}(a\|\theta\|^2 - b)^2$ and is sub-quadratic near its minimum. The GD dynamics near the minimum is given by a cubic map, which is known to show the period-doubling route to chaos Rogers & Whitley (1983). Chen et al. (2023) showed a similar route to chaos by considering a two-layer network with quadratic activation, with the last layer vector $v$ fixed through training and each entry set to one. In this model, the loss is sub-quadratic by *construction* $((\|Ux\|^2 - y)^2$ and the dynamics is given by a cubic map.

# 6 PREDICTIONS AND VERIFICATIONS IN REAL-WORLD SCENARIOS

The preceding analysis offers broader insights and predictions for optimization in real-world models. In this section, we study realistic architectures with real and synthetic datasets and examine the extent to which insights from the UV model generalize.

**Experimental Setup:** Consider a network $f(x; \theta)$, with trainable parameters $\theta$, initialized using normal distribution with zero mean and variance $\sigma_w^2$ in appropriate parameterization. In this section, we use the interpolating parameterization with $s \in [0, 1]$ (detailed in Appendix B.2.1), where networks with $s = 0$ are equivalent to networks in SP as width $n$ goes to infinity and those with $s = 1$ are in $\mu$P. The network is trained on a dataset with $P$ examples using MSE loss and GD. The learning rate is scaled as $\eta = c/\lambda_0^H$, where $c$ is the learning rate constant, and $\lambda_0^H$ is the sharpness at initialization. Additional details provided in figure captions and Appendix B.2.

## 6.1 IMPLICATIONS OF INITIALIZATION AND PARAMETERIZATION FOR REAL-WORLD MODELS

The analysis in Section 5.1 unveils crucial insights into the implicit biases of parameterization in real-world networks. Figure 2(a, d) shows that $\mu$P networks begin training in a flat region of the landscape, where gradients point towards increasing sharpness, and approach the zero loss line while maintaining a low sharpness bias. In contrast, networks in NTP (or equivalently SP), characterized as large initializations, experience sharpness reduction during early training and might not approach with a minimal sharpness bias. The agreement of these observations with networks trained on real-world datasets (Figure 1) suggests that these inherent biases hold in practical scenarios.

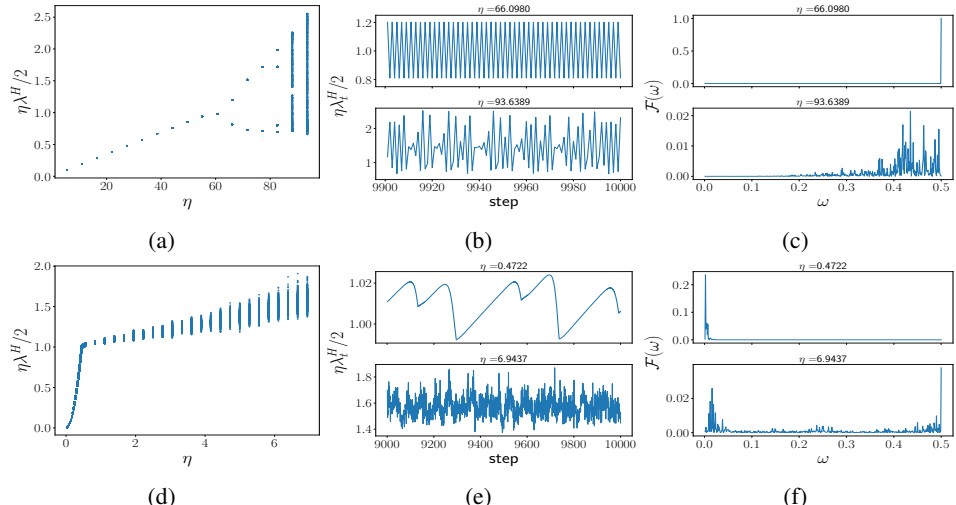

Figure 5: 2-layer linear FCNs trained on (first row) $5,000$ iid random examples with unit output dimension and (second row) $5,000$ CIFAR-10 examples. Different columns correspond to the bifurcation diagram, late-time sharpness trajectories, and the power spectrum of sharpness trajectories. The power spectrum is computed using the last 1000 steps of the trajectories.

## 6.2 SHARPNESS & WEIGHT-NORM CORRELATION AND THE ORIGIN OF FOUR REGIMES

Section 5.1 revealed that, for a wide variety of initializations, at early times trajectories move closer to the saddle point II, resulting in an interim decrease in $\lambda$ (also proportional to the weight norm in this case), before eventually increasing. This critical point where all parameters are zero also exists in real-world models. We thus anticipate that in real-world models, the origin of the four training regimes may be related to a similar mechanism. This would predict a decrease in weight norm as training passes near the saddle point, followed by an eventual increase. In Appendix E, we validate this hypothesis. During the sharpness reduction and intermediate saturation regimes, we see a decrease in the weight norm, followed by an increase in the weight norm as the network undergoes progressive sharpening, following the prediction from the UV model.

## 6.3 THE PHASE DIAGRAM OF EDGE OF STABILITY

Result 1 presents a necessary condition for EoS to occur in the UV model: $\lambda_0 < c\|x\|y/\sqrt{n_{\text{eff}}}$. In real-world models, the initial sharpness $\lambda_0^H$ can be controlled using the initial variance of the weights $\sigma_w^2$. Therefore, this result predicts that real-world models with (i) small initial weight variance $\sigma_w^2$, (ii) large interpolating parameter $s$, or (iii) large learning constant $c$ are more likely to exhibit EoS behavior. Figure 4(a, c) show the phase diagram of EoS, validating these predictions. Additional phase diagrams in Appendix F indicate an enhanced tendency for CNNs and ResNets to exhibit EoS. Furthermore, Figure 4(b, d) show the corresponding test accuracy heatmaps for the above experiments, revealing a positive correlation between observing EoS and improved model performance.

## 6.4 ROUTE TO CHAOS AND BIFURCATION DIAGRAMS

The analysis in Section 5.2 unveiled structured fluctuations in $\lambda$ at the EoS, with a period-doubling route to chaos observed as the learning rate is tuned. This motivates us to analyze fluctuations at the EoS in real-world models trained on realistic and synthetic datasets. Figure 5 shows the bifurcation diagram, late-time sharpness trajectories, and power spectrum of sharpness trajectories for a 2-layer linear FCN. In the first row, the model is trained on random synthetic data with $5,000$ iid examples with unit output dimension, whereas, in the second row, on a $5,000$ example subset of CIFAR-10. Similar to the UV model, FCNs trained on random data exhibit a period-doubling route to chaos, as shown in Figure 5(a). By comparison, FCNs trained on CIFAR-10 only show dense bands in the sharpness rather than exhibiting a clear period-doubling route to chaos.

On analyzing the sharpness trajectories at EoS, we observe long-range correlations in time in real datasets, with fluctuations increasing with the learning rate (see Figure 5(e)). By comparison,

sharpness trajectories of models trained on random datasets exhibit short-period oscillations (see Figure 5(b)). The power spectrum of these sharpness trajectories further quantifies these observations, as shown in Figure 5(c, f). In the random dataset case, high-frequency modes corresponding to the period-doubling route to chaos emerge at EoS as shown in Figure 5(c). In contrast, real datasets exhibit low-frequency modes at small learning rates. As the learning rate is increased, high-frequency modes, reminiscent of the period-doubling route to chaos, start emerging (see Figure 5(f)). In Appendix G.1, we demonstrate that CNNs and ResNets trained on image datasets show dense bands of sharpness similar to those in FCNs.

To understand when the period-doubling route to chaos arises, we perform further analysis in Appendix G. A key determining feature appears to be whether the singular value spectrum of the input-input and output-input covariance matrices are flat or have power-law decay. In Appendix G.2, we show that a 2-layer FCNs trained on a random dataset with power-law singular value spectrum in the input exhibits dense sharpness bands. In Appendix G.3 we show that linear FCNs trained on synthetic datasets with random inputs, such as teacher-student settings and generative settings (details in Appendix B.1), exhibit the period-doubling route to chaos. In contrast, non-linear networks trained on these tasks exhibit dense sharpness bands as observed in real datasets. These observations shed some light on the nature of EoS observed in realistic settings. Nevertheless, a complete understanding of sharpness fluctuations at EoS requires a separate detailed examination.

## 7  DISCUSSION

In this work, we analyzed the crucial effect of initializations and parameterizations on the sharpness dynamics of neural networks and characterized conditions under which sharpness phenomena, such as sharpness reduction, progressive sharpening, and EoS, occur during training.

To develop a deeper understanding of these sharpness phenomena, we analyzed the UV model, which exhibits these sharpness trends. Through a fixed point analysis, we uncover the underlying mechanisms behind these complex sharpness phenomena, such as (i) the mechanism behind early sharpness reduction and progressive sharpening, (ii) the required conditions for the edge of stability, (iii) the crucial role of initialization and parameterization, and (iv) a period-doubling route to chaos on the edge of stability manifold as the learning rate is increased. Finally, we demonstrated that various predictions from this simplified model generalize to real-world scenarios.

While the UV model does capture major sharpness phenomena, it does have some limitations. The UV model does not capture the non-monotonic decrease in the training loss at EoS. This is because the UV model dynamics is effectively described by two variables and the EoS condition $\lambda = 2/\eta$ puts another constraint. As a result, the loss also oscillates along with sharpness at EoS. In comparison, real-world neural networks have a large number of degrees of freedom, and the training can still converge in stable eigendirections. Furthermore, the UV model does not exhibit loss catapult behavior for small effective widths ($n_{\text{eff}} = 1$). In such cases, the threshold for loss catapult ($\eta = 2/\lambda_0$) is comparable to the maximum trainable learning rate. Finally, the UV model does not capture long-range correlations in sharpness dynamics observed in realistic datasets. Through extensive experiments, we identified that long-range correlations in the input covariance matrix results in these long-range correlations.

The applicability of the fixed point analysis extends well beyond the UV model and can be employed in settings involving complex architectures and adaptive optimizers. A prerequisite for applying this method is the closure of the dynamical equations describing the model. While this closure requirement is manageable for deep linear networks through additional function space variables, it becomes challenging for non-linear DNNs where such closure with a few variables is difficult. By analyzing the fixed points of such equations in broader classes of models where closure is achievable, we can gain significant insights into their training dynamics, thereby advancing our understanding of non-convex optimization in neural networks.

Various results such as the phase diagram of EoS, the bifurcation diagram, and the late-time sharpness analysis depend on the training time. Nevertheless, we found that training the models longer does not impact the conclusions presented. In Appendix D.2, we show our results are robust for reasonably small batch sizes ($B \approx 512$). For even smaller batch sizes, the dynamics becomes noise-dominated, and separating the inherent dynamics from noise becomes challenging.

## ACKNOWLEDGEMENTS

D.S.K. and M.B. are supported by an NSF CAREER awards (DMR1753240 and DMR-2345644) and T.H. is supported by the NSF CAREER award (DMR-2045181). The authors acknowledge the University of Maryland supercomputing resources (`http://hpcc.umd.edu`) made available for conducting the research reported in this paper. We thank the anonymous reviewers for their valuable feedback in improving this work. We also thank Darshil Doshi, Andrey Gromov, Jeremy Cohen, Alex Damian, Alex Atanasov, and Lorenzo Noci for insightful discussions. D.S.K. performed part of this work at the Institute for Physical Science and Technology at the University of Maryland, College Park.

## REPRODUCIBILITY STATEMENT

We have provided extensive details for reproducing various theoretical and experimental results presented in the paper. Appendix C details the derivations of the theoretical results for the UV model, while Appendix B provides extensive details for reproducing the presented experiments. Furthermore, the Supplementary materials include the JAX code required to reproduce key results.

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

## A  FURTHER DISCUSSION ON RELATED WORKS

Lewkowycz et al. (2020) examined the early training dynamics of wide networks at large learning rates. Using the top eigenvalue of the Neural Tangent Kernel (NTK) $\lambda^K$ at initialization ($t = 0$), they revealed a 'catapult phase', $2/\lambda_0^K < \eta < \eta_{\max}$, in which training converges despite an initial spike in training loss. Kalra & Barkeshli (2023) analyzed early training dynamics for arbitrary depths and

width and revealed a 'sharpness reduction phase', $2/\lambda_0^H < \eta < c_{\text{loss}}/\lambda_0^H$, which opens up significantly as $c_{\text{loss}}$ increases with depth and $1/$width.

Beyond early training, sharpness continues to increase, until it reaches a break-even point (Jastrzebski et al., 2020), beyond which GD dynamics typically enters the EoS regime (Cohen et al., 2021). This has motivated various theoretical studies to understand GD dynamics at large learning rates: Arora et al. (2022); Rosca et al. (2023); Zhu et al. (2023); Wu et al. (2023); Chen & Bruna (2023); Ahn et al. (2022); Kreisler et al. (2023); Song & Yun (2023); Chen et al. (2023). In particular, Ma et al. Ma et al. (2022) showed that loss functions with sub-quadratic growth exhibit EoS behavior. Arora et al. (2022) show that normalized gradient descent reaches the EoS regime. Damian et al. (2023) analyze the dynamics of the cubic approximation of the loss. Assuming a negative correlation between the gradient direction and the top eigenvector of Hessian, they show that gradient descent dynamics enters a stable cycle in the EoS regime. Ahn et al. (2022) analyze EoS in a single-neuron 2-layer network and a simplified three-parameter ReLU network assuming the existence of a 'forward invariant subset' near the minima. Kreisler et al. (2023) analyzed scalar linear networks to show that the sharpness attained by the gradient flow dynamics monotonically decreases in the EoS regime. Wu et al. (2023) demonstrate that gradient descent, with any learning rate in the EoS regime, optimizes logistic regression with linearly separated data over large time scales. Below, we discuss closely related works in detail and clarify the distinction with our work.

Wang et al. (2022) analyze EoS in a 2-layer linear network using the norm of the last layer. They solely focus on cases that exhibit progressive sharpening right from initializations by considering assumptions (refer to Assumptions 4.1 and 4.2 of their paper) on the training dataset. Contrary to these assumptions, Figure 1(e) demonstrates that such assumptions are invalid in many realistic settings, where progressive sharpening is not observed at all.

Agarwala et al. (2022) showed that a modified model exhibits progressive sharpening and two-step oscillations at EoS using NTK as the proxy. They state that the UV model does not exhibit EoS behavior (see Section 3.2.1 of the referenced paper). This is because their analysis is restricted to the Standard Parmaeterization corresponding to Figure 2(c, f) (c.f. Figure 8 of Agarwala et al. (2022)). In contrast, we show that the UV model exhibits EoS behavior under the appropriate choice of parameterization and training example.

Zhu et al. (2023) proved EoS convergence for the loss $\frac{1}{4}(x^2y^2 - 1)^2$, where $x, y \in \mathbb{R}$. Additionally, they empirically demonstrated a bifurcation diagram in the space of abstract variables of $x$ and $y$. It is worth noting that while these bifurcations arise from the same underlying behavior, they contrast with our route to chaos bifurcation diagrams which quantify sharpness fluctuations with learning rate.

Chen & Bruna (2023) analyze two-step gradient updates of a single-neuron network and matrix factorization to gain insights into EoS. Similar to our work, they show a bifurcation diagram of sharpness against the learning rate for the matrix factorization problem. While the scalar matrix factorization problem can be mapped to the UV model with a specific choice of $\frac{\|x\|}{\sqrt{n_{\text{eff}}}}$, it is not straightforward to apply their conclusions to the neural network setting, as it requires the correct choice of parameterization. In particular, the UV model under NTP parameterization, as shown in Figure 2(c, f), does not display EoS behavior at considerable widths, a finding also noted by Agarwala et al. (2022). Observing EoS requires the correct choice of the parameterization ($\mu$P) and training example. Furthermore, although the scalar matrix factorization in Chen & Bruna (2023) can be mapped to a special case of the UV model considered in our work, we provide significant additional insights. In particular, with respect to the bifurcation phenomena in the UV model, we explain the existence of an attractor submanifold on which the EoS behavior occurs. We further show that on the EoS submanifold, the loss becomes subquadratic in nature and the gradient descent dynamics therefore become approximated by the cubic map, which is well-studied in the chaos literature. This makes clear the origin of the period-doubling route to chaos. Additionally, in Section 6, we extend the analysis of EoS beyond the UV model, comparing sharpness trajectories of synthetic and real datasets at EoS. In contrast to the synthetic datasets, sharpness trajectories of real datasets show long-range correlations in time. We take the first steps by attributing these long-range correlations to correlations in the dataset. Note that this setting cannot be mapped to the matrix factorization setting.

Song & Yun (2023) show that late-time trajectories oscillate around $2f/\eta\ell'$, where $f$ is the network output and $\ell'$ is the derivative of the loss. They refer to the term bifurcation diagram to describe these phenomena, contrasting with the sharpness versus learning rate bifurcation diagrams presented in our

study. We quote Section 3 from their paper ''..we plot the bifurcation diagram $q = r(p) = \ell'(p)/p$ and observe that GD trajectories tend to align with this curve..'' Here, $p$ and $q$ correspond to $\Delta f$ and $\frac{2}{\eta\lambda}$ in our setting. They plot trajectories in the $(p, q) \equiv (\Delta f, \frac{2}{\eta\lambda})$ plane and their condition $q = r(p)$ simply corresponds to the EoS condition $\lambda = \frac{2}{\eta}$ for MSE loss. In contrast, our work presents bifurcation diagrams resulting from how the sharpness fluctuations vary with the learning rate. Therefore, the bifurcation diagrams from these works are not directly related to the route-to-chaos bifurcation diagrams presented in our work.

Kong & Tao (2020) were the first ones to show that gradient descent dynamics becomes chaotic at large learning rates and converges to a statistical distribution instead of a minimum.

Chen et al. (2023) analyzed large learning rate dynamics of toy models which are characterized by a one-dimensional cubic map and demonstrated five different training phases: (a) monotonic, (b) catapult, (c) periodic, (d) chaotic, and (e) divergent. In particular, they considered a two-layer network with quadratic activation, where the last layer vector $\mathbf{v}$ is not trained and each entry is set to one. This model belongs to a family that is effectively described by one variable $\Delta f$. In this model, the loss is sub-quadratic by construction $(||Ux||^2 - y)^2$. In contrast, the UV model that we study is an effectively two-variable model and in these cases, training dynamically finds the attractive EoS manifold such that the loss has a sub-quadratic nature on this submanifold.

Concurrent work by Wang et al. (2023) categorizes training trajectories into three stages: (i) sharpness reduction, (ii) progressive sharpening, and (iii) edge of stability. They argue that different large learning rate behavior depends on the 'regularity' of the loss landscape. Specifically, they generalize toy landscapes from existing studies with parameters controlling the regularity. They show that models with good regularity first experience a decrease in sharpness and then progressive sharpening and enter the edge of stability.

Noci et al. (2024) examined the sharpness dynamics of networks with parameterization. They argue that the learning rate transfer property of $\mu$P is correlated with consistent sharpness trajectories across varying depths and widths.

# B  EXPERIMENTAL DETAILS

## B.1  DATASETS

**Standard image datasets:**  We considered the MNIST Deng (2012), Fashion-MNIST Xiao et al. (2017), and CIFAR-10 Krizhevsky (2009) datasets. The images are standardized to have zero mean and unit variance across the feature dimensions, and target labels are represented as one-hot encodings.

**Random dataset:**  We construct a random dataset $(X, Y) = \{(\boldsymbol{x}^\mu, \boldsymbol{y}^\mu)\}_{\mu=1}^P$ with $\boldsymbol{x}^\mu \sim \mathcal{N}(0, I)$ and $\boldsymbol{y}^\mu \sim \mathcal{N}(0, I)$, both sampled independently. Note there is no correlation between inputs and outputs.

**Teacher-student dataset:**  Consider a teacher network $f(\boldsymbol{x}; \theta_0)$ with $\theta_0$ initialized randomly as described in Appendix B.2. Then, we construct a teacher-student dataset $(X, Y) = \{(\boldsymbol{x}^\mu, \boldsymbol{y}^\mu)\}_{\mu=1}^P$ with $\boldsymbol{x}^\mu \sim \mathcal{N}(0, I)$ and $\boldsymbol{y}^\mu = f(\boldsymbol{x}^\mu; \theta_0)$.

**Random power-law dataset:**  Starting with the random dataset $(X', Y')$, we utilize the singular value decomposition of the input and output matrices

$$X' = P_x S_{x'} Q_x^T, \qquad\qquad Y' = P_y S_{y'} Q_y^T. \qquad (4)$$

Next, we rescale the $k^{th}$ singular value of $S_{x'}$ and $S_{y'}$ as

$$(S_x)_k = A_x (S_{x'})_k k^{-B_x} \qquad\qquad (S_y)_k = A_y (S_{y'})_k k^{-B_y}, \qquad (5)$$

and re-construct input and output matrices as below

$$X = P_x S_x Q_x^T, \qquad\qquad Y = P_y S_y Q_y^T. \qquad (6)$$

The variables $A_x, B_x, A_y$, and $B_y$ uniquely characterize the dataset.

**Generative image dataset:** Given a pre-trained network $f(\boldsymbol{x}; \theta)$ on a standard image dataset listed above, we construct a generative image dataset $(X, Y) = \{(\boldsymbol{x}^\mu, \boldsymbol{y}^\mu)\}_{\mu=1}^P$ with $\boldsymbol{x}^\mu \sim \mathcal{N}(0, I)$ and $\boldsymbol{y}^\mu = f(\boldsymbol{x}^\mu; \theta)$.

## B.2 MODELS

**FCNs:** We considered ReLU FCNs without bias with uniform hidden layer width $n$.

**CNNs:** We considered Myrtle family ReLU CNNs Shankar et al. (2020) without any bias with a fixed number of channels in each layer, which we refer to as the width of the network.

**ResNets:** We adapted ResNet He et al. (2016) implementations from Flax examples. Our implementation uses Layer norm and initialize the weights as $\mathcal{N}(0, \sigma_w^2/\text{fan}_{\text{in}})$. For ResNets, we refer to the number of channels in the first block as the width.

We implemented all models using the JAX Bradbury et al. (2018), and Flax libraries Heek et al. (2020).

### B.2.1 DETAILS OF NETWORK PARAMETERIZATION

In this section, we describe different parameterizations used in the paper. For simplicity, we describe the parameterizations for FCNs. Nevertheless, these arguments generalize to other architectures.

**Standard Parameterization (SP):** Consider a neural network $f : \mathbb{R}^{d_{\text{in}}} \to \mathbb{R}^{d_{\text{out}}}$ with $d$ layers and constant width $n$. Then, standard parameterization is defined as follows:

$$\begin{aligned}
\boldsymbol{h}^{(1)}(\boldsymbol{x}) &= W^{(1)}\boldsymbol{x}, \\
\boldsymbol{h}^{(l+1)}(\boldsymbol{x}) &= W^{(l+1)}\phi\left(\boldsymbol{h}^{(l)}(\boldsymbol{x})\right), \\
\boldsymbol{f}(\boldsymbol{x}; \theta) &= W^{(d)}\phi\left(\boldsymbol{h}^{(d-1)}(\boldsymbol{x})\right),
\end{aligned} \qquad (7)$$

where $W^{(1)} \sim \mathcal{N}(0, \sigma_w^2/d_{\text{in}})$, $W^{(l)} \sim \mathcal{N}(0, \sigma_w^2/n)$ for $1 < l < d$, and $W^{(d)} \sim \mathcal{N}(0, 1/n)$; $\phi(\cdot)$ is the elementwise activation function. The input is normalized such that $\|\boldsymbol{x}\|^2 = d_{\text{in}}$.

**Neural Tangent Parameterization (NTP):** Consider a neural network $f : \mathbb{R}^{d_{\text{in}}} \to \mathbb{R}^{d_{\text{out}}}$ with $d$ layers and constant width $n$. Then, the Neural Tangent Parameterization is defined as follows:

$$\begin{aligned}
\boldsymbol{h}^{(1)}(\boldsymbol{x}) &= \frac{\sigma_w}{\sqrt{d_{\text{in}}}}W^{(1)}\boldsymbol{x}, \\
\boldsymbol{h}^{(l+1)}(\boldsymbol{x}) &= \frac{\sigma_w}{\sqrt{n}}W^{(l+1)}\phi\left(\boldsymbol{h}^{(l)}(\boldsymbol{x})\right), \\
\boldsymbol{f}(\boldsymbol{x}; \theta) &= \frac{1}{\sqrt{n}}W^{(d)}\phi\left(\boldsymbol{h}^{(d-1)}(\boldsymbol{x})\right),
\end{aligned} \qquad (8)$$

where $W^{(l)} \sim \mathcal{N}(0, 1)$ for $1 \le l \le d$ and $\phi(\cdot)$ is the elementwise activation function. The input is normalized such that $\|\boldsymbol{x}\|^2 = d_{\text{in}}$.

Both SP and NTP are closely related parameterizations —SP with learning rate $\eta = \Theta(1/n)$ learning rate becomes equivalent to NTP when the input dimension is equal to the width ($d_{\text{in}} = n$) Yang & Hu (2021).

**Interpolating Parameterization:** Consider a neural network $f : \mathbb{R}^{d_{in}} \to \mathbb{R}^{d_{out}}$ with $d$ layers and constant width $n$. Let $W^{(l)}$ denote the weight matrix at layer $l$. Then, "interpolating parameterization" is defined as follows:

$$
\begin{aligned}
\boldsymbol{h}^{(1)}(\boldsymbol{x}) &= n^{s/2} W^{(1)} \boldsymbol{x}, \\
\boldsymbol{h}^{(l+1)}(\boldsymbol{x}) &= W^{(l+1)} \phi\left(\boldsymbol{h}^{(l)}(\boldsymbol{x})\right), \\
\boldsymbol{f}(\boldsymbol{x}; \theta) &= \frac{1}{n^{s/2}} W^{(d)} \phi\left(\boldsymbol{h}^{(d-1)}(\boldsymbol{x})\right),
\end{aligned}
\tag{9}
$$

Here, $s$ is a parameter that interpolates between standard-like parameterization and maximal update parameterization. The weight matrices are sampled from Gaussian distributions: $W^{(1)} \sim \mathcal{N}(0, \sigma_w^2/n^s)$, $W^{(l)} \sim \mathcal{N}(0, \sigma_w^2/n)$ for $1 < l < d$, and $W^{(d)} \sim \mathcal{N}(0, 1/n)$. We normalize the input such that $\|\boldsymbol{x}\| = 1$.

**Maximal update Parameterization ($\mu$P):** The maximal update parameterization corresponds to the $s = 1$ case in the above setting.

### B.3 DETAILS OF FIGURES

**Figure 1:** Training loss and sharpness trajectories of 4-layer ReLU FCNs with $n = 512$, trained on a subset of $5,000$ CIFAR-10 examples using MSE loss and GD: (a, d) SP with $\sigma_w^2 = 0.5$, (b, e) SP with $\sigma_w^2 = 2.0$, (c, f) $\mu P$ with $\sigma_w^2 = 2.0$.

**Figure 2** Training trajectories of the UV model with $\|\boldsymbol{x}\| = 1$ and $y = 2$ in the $(\Delta f, \lambda)$ plane for different values of $n$, $n_{\text{eff}}$ and $\eta$. The columns show initializations with different $n$ and $n_{\text{eff}}$, while the rows represent increasing learning rates for fixed initializations. The horizontal dash-dot line $\eta\lambda = 2$ separates the stable (solid black vertical line) and unstable (dashed black vertical line) fixed points along the zero loss fixed line I. Forbidden regions, $2\|\boldsymbol{x}\||\Delta f + y|/\sqrt{n_{\text{eff}}} > \lambda$, (see Appendix C.2) are shaded gray. The nullclines $\Delta f_{t+1} = \Delta f_t$ and $\lambda_{t+1} = \lambda_t$ are shown as orange and white dashed curves, respectively. Sharpness reduction, progressive sharpening, and divergent regions are colored green, yellow, and blue. The gray arrows indicate the local vector field $\hat{G}(\Delta f, \lambda)$, which is the direction of the updates. The training trajectories are depicted as black lines with arrows, with the star marking the initialization. In all cases, $\eta_c = \sqrt{n_{\text{eff}}}/2$ (introduced in Section 5.2).

**Figure 3:** *UV model dynamics on the EoS manifold*:(a) Bifurcation diagram depicting late-time limiting values of $\lambda$ obtained by simulating Equation (3). (b) Bifurcation diagram of the UV model. In both figures, $\|\boldsymbol{x}\| = 1, y = 2$ and $n_{\text{eff}} = 1$ and $\eta_c = 0.5$.

**Figure 4:** *Phase diagram of EoS*: (a) Heatmap of $\eta\bar{\lambda}^H/2$ of 3-layer ReLU FCNs with $s = 0$ trained on a subset of $5,000$ CIFAR-10 examples for 10k steps, with the weight variance $\sigma_w^2$ and learning rate multiplier $c = \eta\lambda_0^H$ as axes. $\bar{\lambda}^H$ is obtained by averaging $\lambda_t^H$ over last 200 steps. As the color varies from blue to white, $\eta\bar{\lambda}^H/2$ increases, where the brightest white region indicates the EoS regime with $\eta\bar{\lambda}^H/2 \geq 1$. (b) Same heatmap with fixed $\sigma_w^2 = 2.0$, but varying $s$ continuously.

**Figure 5:** *EoS in synthetic vs real-datasets:* 2-layer linear FCN trained on (first row) $5,000$ iid random examples with unit output dimension and (second row) $5,000$ CIFAR-10 examples. Different columns correspond to the bifurcation diagram, late-time sharpness trajectories, and the power spectrum of sharpness trajectories. Both models are trained for 10k steps using GD.

**Figure 9:** *Two-step phase portrait of UV model in $(\Delta f, \beta)$ phase plane*: These plots are equivalent to Figure 2(d-f), but with training trajectory and local are plotted for every other step.

**Figure 15:** Sharpness and Weight Norm of 3-layer ReLU FCNs in SP with $\sigma_w^2 = 1/3$ and width 200, trained on a subset of CIFAR-10 with $5,000$ examples using GD.

### B.4 SHARPNESS MEASUREMENT

We measure sharpness using the power iteration method with $m$ iterations. Typically, $m = 20$ iterations ensure convergence. Exceptions requiring more iterates are discussed separately.

### B.5 POWER SPECTRUM ANALYSIS

For a given signal $x'(t)$, we standardize the signal

$$x(t) = \frac{x'(t) - \mu}{\sigma}, \tag{10}$$

where $\mu$ is the mean and $\sigma^2$ is the variance of the signal. Subtracting the mean removes the zero frequency component in the power spectrum. Next, consider the discrete Fourier transform $\mathcal{F}(\omega)$ of $x(t)$:

$$\mathcal{F}(\omega) = \frac{1}{T} \sum_{t=0}^{T-1} e^{-i2\pi\omega t/T} x(t), \tag{11}$$

Then, the power spectrum is $P(\omega) = |\mathcal{F}(\omega)|^2$. The normalization by $T$ in the Fourier transform ensures that the sum of the power spectrum is equal to the variance of the signal, i.e., $\sum_\omega P(\omega) = \sigma^2$.

### B.6 ESTIMATION OF COMPUTATIONAL RESOURCES USED

Most of our experiments, aside from the phase diagrams, required minimal computational resources, estimated to be less than 50 A100 hours. In contrast, each phase diagram required 50 A100 hours, totaling 500 A100 hours for all phase diagrams. Including initial experiments, we expect our total usage to be under 600 A100 hours.

## C PROPERTIES OF THE UV MODEL

### C.1 DERIVATION OF THE FUNCTION SPACE DYNAMICS

Equations (1) and (2) can be derived using the gradient descent update equations:

$$U_{t+1} = U_t - \eta \frac{\Delta f_t \boldsymbol{v}_t \boldsymbol{x}^T}{\sqrt{n_{\text{eff}}}}, \tag{12}$$

$$\boldsymbol{v}_{t+1} = \boldsymbol{v}_t - \eta \frac{\Delta f_t U_t \boldsymbol{x}_t}{\sqrt{n_{\text{eff}}}}. \tag{13}$$

At step $t + 1$, the residual $\Delta f_{t+1}$ can be written in terms of the gradient updates of $U$ and $\boldsymbol{v}$:

$$\Delta f_{t+1} = f_{t+1} - y \tag{14}$$

$$= \frac{1}{\sqrt{n_{\text{eff}}}} \boldsymbol{v}_{t+1}^T U_{t+1} \boldsymbol{x} - y \tag{15}$$

$$= \frac{1}{\sqrt{n_{\text{eff}}}} \left( \boldsymbol{v}_t - \eta \frac{\Delta f_t U_t \boldsymbol{x}}{\sqrt{n_{\text{eff}}}} \right)^T \left( U_t - \eta \frac{\Delta f_t \boldsymbol{v}_t \boldsymbol{x}^T}{\sqrt{n_{\text{eff}}}} \right) \boldsymbol{x} - y \tag{16}$$

$$= \Delta f_t - \frac{\eta \Delta f_t}{n_{\text{eff}}} \left( \boldsymbol{x}^T U_t^T U_t \boldsymbol{x} + \boldsymbol{v}_t^T \boldsymbol{v}_t \boldsymbol{x}^T \boldsymbol{x} \right) + \frac{\eta^2 \|\boldsymbol{x}\|^2 \Delta f_t^2}{n_{\text{eff}}} \left( \frac{1}{\sqrt{n_{\text{eff}}}} \boldsymbol{x}^T U_t^T \boldsymbol{v}_t \right) \tag{17}$$

$$= \Delta f_t \left( 1 - \eta \lambda_t + \frac{\eta^2 \|\boldsymbol{x}\|^2}{n_{\text{eff}}} \Delta f_t (\Delta f_t + y) \right). \tag{18}$$

Here, $\Delta f_{t+1}$ only depends on $\Delta f_t$ and $\lambda_t$. Similarly, we write down the $\lambda_{t+1}$ using the gradient update equations:

$$\lambda_{t+1} = \frac{1}{n_{\text{eff}}} \left( \boldsymbol{x}^T U_{t+1}^T U_{t+1} \boldsymbol{x} + \boldsymbol{v}_{t+1}^T \boldsymbol{v}_{t+1} \boldsymbol{x}^T \boldsymbol{x} \right) \tag{19}$$

$$= \lambda_t - 4 \frac{\eta \|\boldsymbol{x}\|^2}{n_{\text{eff}}} \Delta f_t (\Delta f_t + y) + \frac{\eta^2 \|\boldsymbol{x}\|^2 \Delta f_t^2}{n_{\text{eff}}} \lambda_t \tag{20}$$

$$= \lambda_t + \frac{\eta \|\boldsymbol{x}\|^2}{n_{\text{eff}}} \Delta f_t^2 \left( \eta \lambda_t - 4 \frac{\Delta f_t + y}{\Delta f_t} \right). \tag{21}$$

Equations (18) and (21) form a closed system. This means that $\Delta f_{t+1}$ and $\lambda_{t+1}$ are completely described using $\Delta f_t$ and $\lambda_t$. As a result, the complete dynamics of the UV model can be fully described using only these two variables with three parameters effective parameters $\eta$, $\frac{\|x\|^2}{n_{\text{eff}}}$ and $y$.

### C.2 FORBIDDEN REGIONS OF THE UV MODEL

In this section, we utilize the non-negativity of $\lambda$ to derive the condition for allowed regions within the phase plane for the UV model. Consider the function space equations written in terms of the pre-activation $\boldsymbol{h}(\boldsymbol{x}) = U\boldsymbol{x}$:

$$f(\boldsymbol{x}; \theta) = \frac{1}{\sqrt{n_{\text{eff}}}} \boldsymbol{v}^T \boldsymbol{h}(\boldsymbol{x}) \tag{22}$$

$$\lambda = \frac{1}{n_{\text{eff}}} \left( \|\boldsymbol{v}\|^2 \|\boldsymbol{x}\|^2 + \|\boldsymbol{h}\|^2 \right). \tag{23}$$

Let $\cos(\boldsymbol{h}, \boldsymbol{v})$ denote the cosine similarity between $\boldsymbol{v}$ and $\boldsymbol{h}$. Then, the network output is bounded as

$$|\cos(\boldsymbol{h}, \boldsymbol{v})| = \frac{\sqrt{n_{\text{eff}}} |\Delta f + y|}{\|\boldsymbol{v}\| \|\boldsymbol{h}\|} \leq 1 \tag{24}$$

Next, using $(\|\boldsymbol{v}\| \|\boldsymbol{x}\| - \|\boldsymbol{h}\|)^2 \geq 0$, we can bound the product $\|\boldsymbol{v}\| \|\boldsymbol{h}\|$ using $\lambda$

$$\frac{2\|\boldsymbol{x}\|}{\sqrt{n_{\text{eff}}}} |\Delta f + y| \leq \lambda. \tag{25}$$

The derived inequality describes the allowed phase plane regions for the UV model.

### C.3 FIXED POINTS AND LINE

To identify the fixed points, we set $\Delta f_{t+1} = \Delta f_t$ and $\lambda_{t+1} = \lambda_t$ in Equations (1) and (2). This yields the dynamical fixed points of the UV model. Table 1 lists these fixed points along with their stability. Additionally, it provides the eigenvalues and eigenvectors of the Jacobian for the update maps described by Equations (1) and (2), evaluated at the fixed points.

### C.4 THE MAXIMUM LEARNING RATE $\eta_{\text{UPPER}}$

In Section 4, we stated that for $\eta > \eta_{\text{upper}} = 2\eta_c$, training diverges for all initializations except for those at the fixed points. Here, we justify this claim.

First, Figure 6 shows that as $\eta$ approaches $\eta_{\text{upper}}$, fixed point III merges with fixed point II, reducing the convergence region to the EoS manifold. At this learning rate, the stability of fixed point II changes from saddle to unstable as the corresponding eigenvalue $(1 - \frac{\eta}{\eta_c})^2 \big|_{\eta=2\eta_c}$ surpasses 1. Consequently, any initialization outside the EoS manifold results in divergence. Next, Figure 3(a) shows that on the EoS manifold training diverges for $\eta > \eta_{\text{upper}}$. This corroborates our initial claim.

| | $(\Delta f^*, \lambda^*)$ | eigenvalues | eigenvectors | Linear stability |
|---|---|---|---|---|
| I | $(0, \lambda)$ for $\lambda \geq \frac{2\|x\|y}{\sqrt{n_{\text{eff}}}}$ | $1, 1 - \eta\lambda$ | $\begin{bmatrix}0\\1\end{bmatrix}, \begin{bmatrix}\frac{n_{\text{eff}}\lambda}{4\|x\|^2 y}\\1\end{bmatrix}$ | $\begin{cases}\text{stable} & \eta\lambda < 2\\ \text{unstable} & \eta\lambda > 2\end{cases}$ |
| II | $(-y, 0)$ | $(1 - \frac{\eta\|x\|y}{\sqrt{n_{\text{eff}}}})^2, (1 + \frac{\eta\|x\|y}{\sqrt{n_{\text{eff}}}})^2$ | $\begin{bmatrix}-\frac{\sqrt{n_{\text{eff}}}}{2\|x\|}\\1\end{bmatrix}, \begin{bmatrix}\frac{\sqrt{n_{\text{eff}}}}{2\|x\|}\\1\end{bmatrix}$ | saddle |
| III | $\left(\frac{-2\sqrt{n_{\text{eff}}}}{\|x\|\eta}, \frac{4}{\eta} - \frac{2\|x\|y}{\sqrt{n_{\text{eff}}}}\right)$ | $9, 5 - \frac{2\eta\|x\|y}{\sqrt{n_{\text{eff}}}}$ | $\begin{bmatrix}\frac{n_{\text{eff}}}{\eta\|x\|^2 y}\\1\end{bmatrix}, \begin{bmatrix}-\frac{\sqrt{n_{\text{eff}}}}{2\|x\|}\\1\end{bmatrix}$ | unstable |
| IV | $\left(\frac{2\sqrt{n_{\text{eff}}}}{\|x\|\eta}, \frac{4}{\eta} + \frac{2\|x\|y}{\sqrt{n_{\text{eff}}}}\right)$ | $9, 5 + \frac{2\eta\|x\|y}{\sqrt{n_{\text{eff}}}}$ | $\begin{bmatrix}\frac{n_{\text{eff}}}{\eta\|x\|^2 y}\\1\end{bmatrix}, \begin{bmatrix}\frac{\sqrt{n_{\text{eff}}}}{2\|x\|}\\1\end{bmatrix}$ | unstable |

Table 1: Fixed line (I) and points (II-IV) and corresponding eigenvalues and eigenvectors of the Jacobian of the update map in Equations (1) and (2). The stability is determined for $\eta < \eta_{\text{upper}} = 2\eta_c = 2\sqrt{n_{\text{eff}}}/\|x\|y$. Above this threshold, training diverges for all initializations except for those at the fixed points, as demonstrated in Appendix C.4.

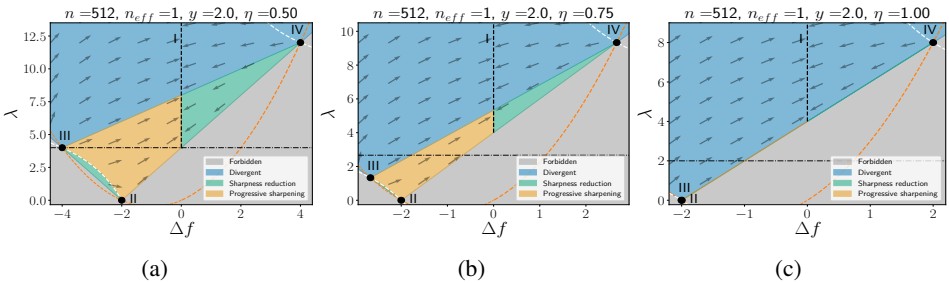

(a)        (b)        (c)

Figure 6: Phase portrait of the UV model for different learning rates $\eta$. The critical learning rate is $\eta_c = 0.5$ and the maximum learning rate is $\eta_{\text{upper}} = 1.0$.

## C.5 SHARPNESS VERSUS THE TRACE OF HESSIAN

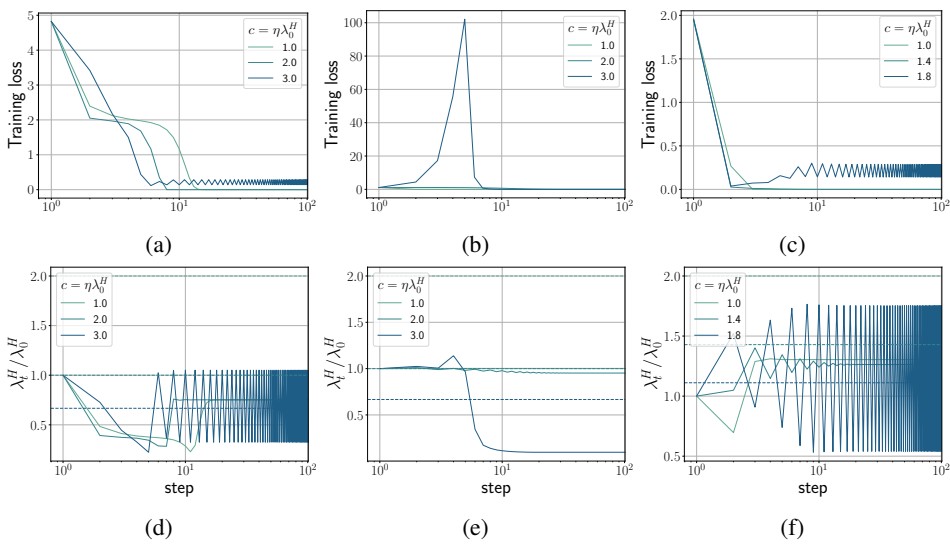

(a)        (b)        (c)

(d)        (e)        (f)

Figure 7: Training trajectories of the UV model trained on a single example with $\|x\| = 1$ and $y = 2$ using MSE loss and GD: (a, d) NTP with $n = 1$, $\sigma_w^2 = 0.5$, (b, e) NTP with $n = 512$, $\sigma_w^2 = 1.0$, and (c, f) $\mu$P with $n = 512$, $\sigma_w^2 = 1.0$.

In this section, we show that the trace of the Hessian $\lambda$ (which is also the scalar NTK in this case), is an adequate proxy for sharpness. Figure 8 shows training trajectories of the UV model, with $\lambda$ as a proxy for sharpness and learning rate scaled as $\eta = k/\lambda_0$. These $\lambda$ trajectories show similar trends to

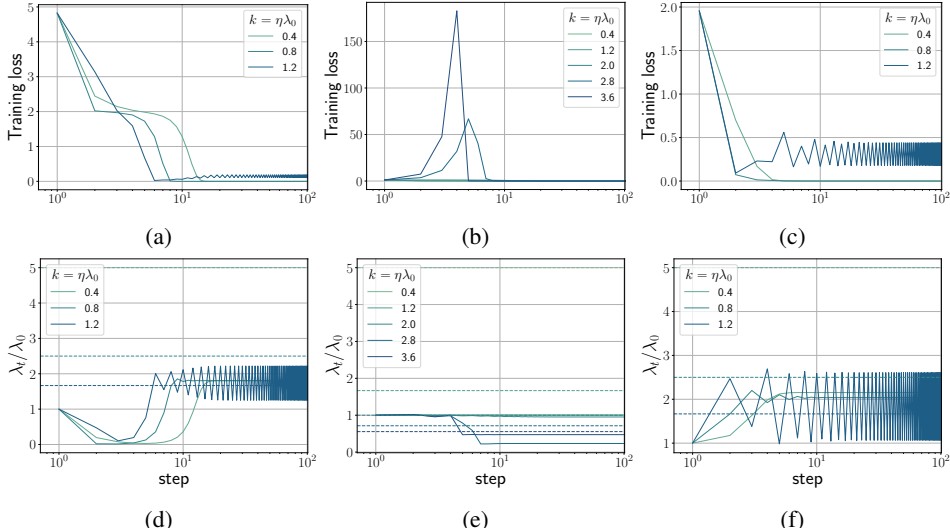

Figure 8: *UV model shows all four training regimes:* Training trajectories of the UV model trained on a single example $(\boldsymbol{x}, y)$ with $\|\boldsymbol{x}\| = 1$ and $y = 2$ using MSE loss and gradient: (a, d) NTP with $n = 1$ and $\sigma_w^2 = 0.5$ (b, e) NTP with $n = 512$ and $\sigma_w^2 = 1.0$, and (c, f) $\mu$P with $n = 1$ and $\sigma_w^2 = 1.0$.

those of $\lambda^H$ observed in Figure 7, with one key difference: during early training, $\lambda$ does not catapult during early training at large widths (compare Figure 7(e) and Figure 8(e)). Otherwise, $\lambda$ effectively captures other qualitative behavior of $\lambda^H$.

### C.6   THE DISTRIBUTION OF RESIDUAL AND NTK AT INITIALIZATION

In this section, we compute the distribution of $\Delta f$ and $\lambda$ for the UV model at initialization. Consider the UV model written in terms of the pre-activation $h(x) = Ux$,

$$f(\boldsymbol{x}; \theta) = \frac{1}{\sqrt{n^{1-p}}} \boldsymbol{v}^T \boldsymbol{h}(\boldsymbol{x}) \tag{26}$$

$$\lambda = \frac{1}{n^{1-p}} \left( \|\boldsymbol{v}\|^2 \|\boldsymbol{x}\|^2 + \|\boldsymbol{h}\|^2 \right), \tag{27}$$

with $v_i, U_{ij} \sim \mathcal{N}(0, \sigma_w^2/n^p)$. Then, each pre-activation $h_i$ is normally distributed at initialization with zero mean and variance

$$\mathbb{E}_\theta[h_i^2] = \sum_{j,k=1}^{d_{\text{in}}} \langle U_{ij} U_{ik} \rangle x_j x_k = \sum_{j,k=1}^{d_{\text{in}}} \frac{\sigma_w^2}{n^p} \delta_{jk} x_j x_k = \frac{\sigma_w^2 \|\boldsymbol{x}\|^2}{n^p}. \tag{28}$$

Hence, each pre-activation is distributed as $h_i \sim \mathcal{N}(0, \sigma_w^2 \|\boldsymbol{x}\|^2/n^p)$. It follows that the network output is also normally distributed at initialization with zero mean and variance

$$\mathbb{E}_\theta[f_0^2] = \frac{1}{n^{1-p}} \sum_{i,j=1}^{n} \langle v_i v_j \rangle \langle h_i h_j \rangle = \frac{1}{n^{1-p}} \sum_{i=1}^{n} \frac{\sigma_w^2}{n^p} \frac{\sigma_w^2 \|\boldsymbol{x}\|^2}{n^p} = \frac{\sigma_w^4 \|\boldsymbol{x}\|^2}{n^p}. \tag{29}$$

Hence, the residual at initialization is distributed as $\Delta f_0 \sim \mathcal{N}\left(-y, \sigma_w^4 \|\boldsymbol{x}\|^2/n^p\right)$. Similarly, we can also compute the distribution of $\lambda$ at initialization. The mean value of $\lambda$ is given by

$$\mathbb{E}_\theta[\lambda_0] = \frac{1}{n^{1-p}}\left(\|\boldsymbol{x}\|^2\langle\|\boldsymbol{v}\|^2\rangle + \langle\|\boldsymbol{h}\|^2\rangle\right) = 2\sigma_w^2\|\boldsymbol{x}\|^2, \tag{30}$$

where we have used $\langle\|\boldsymbol{v}\|^2\rangle = \sigma_w^2 n^{1-p}$ and $\langle\|\boldsymbol{h}\|^2\rangle = \sigma_w^2\|\boldsymbol{x}\|^2 n^{1-p}$. Using similar computations, the second moment of $\lambda$ is given by:

$$\mathbb{E}_\theta[\lambda_0^2] = \frac{1}{n^{2-2p}}\left(\|\boldsymbol{x}\|^4\langle\|\boldsymbol{v}\|^4\rangle + \langle\|\boldsymbol{h}\|^4\rangle + 2\|\boldsymbol{x}\|^2\langle\|\boldsymbol{v}\|^2\|\boldsymbol{h}\|^2\rangle\right) = \frac{4(n+1)}{n}\sigma_w^4\|\boldsymbol{x}\|^4. \tag{31}$$

Hence, the $\lambda$ at initialization is distributed as $\lambda_0 \sim \mathcal{N}\left(2\sigma_w^2\|\boldsymbol{x}\|^2, 4\sigma_w^4\|\boldsymbol{x}\|^4/n\right)$.

### C.7 EoS MANIFOLD IS A DYNAMICAL ATTRACTOR

To demonstrate that late time trajectories for $\eta > \eta_c$ converge to the EoS manifold, we define $\beta := \frac{\sqrt{n_{\text{eff}}}}{2\|\boldsymbol{x}\|}\lambda - (\Delta f + y)$. $\beta$ lies on the direction orthogonal to the EoS manifold, such that $\beta = 0$ corresponds to the manifold itself, while $\beta < 0$ is forbidden. Under this transformation, $\beta$ updates as $\beta_{t+1} = \beta_t(1 + \frac{\eta\|\boldsymbol{x}\|\Delta f_t}{\sqrt{n_{\text{eff}}}})^2$. It follows that $\beta^* = 0$ stays invariant under the dynamics and defines a nullcline.

Due to oscillations in $\Delta f$ near convergence, it is instructive to examine the two-step dynamics Agarwala et al. (2022); Chen & Bruna (2023), compactly denoted as $(\Delta f_{t+2}, \lambda_{t+2}) := M^{(2)}(\Delta f_t, \lambda_t)$. Figure 9 shows the two-step trajectories and the corresponding vector field $\hat{G}^{(2)}(\Delta f, \beta)$ in the $(\Delta f, \beta)$ plane.

We observe that there exists a critical $\eta_c$ such that for $\eta < \eta_c$, $\hat{G}^{(2)}(\Delta f, \beta)$ points towards the stable zero-loss line (see Figure 9(a)). By comparison, for $\eta > \eta_c$, all points along the zero-loss line become unstable and the vector field directs towards points on the $\beta = 0$ line, as shown in Figure 9(b). The critical $\eta_c$ for which all points on the zero-loss line become unstable thus gives a necessary condition for EoS:

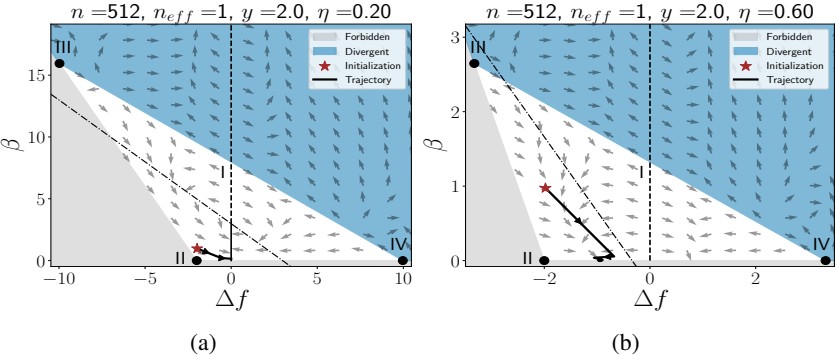

(a)  (b)

Figure 9: These plots are equivalent to Figure 2(a, d) ($\eta_c = 0.5$), but with training trajectory and local vector field plotted for every other step in $(\Delta f, \beta)$ plane. The tilted dash-dotted line indicates the $\eta\lambda = 2$ line.

### C.8 CRITICAL LEARNING RATE FOR EDGE OF STABILITY

In this section, we estimate the required condition on the learning rate for the UV model to exhibit EoS. We specifically focus on the case with $y > 0$ as for $y = 0$, $\lambda$ can only decrease. As a result, the model does not exhibit progressive sharpening and EoS. In Section 5, we observed that the EoS occurs as the zero-loss minima with the smallest $\lambda$ becomes unstable. From Equation (25) it follows that the smallest $\lambda$ with zero loss is

$$\lambda_{\min} = \frac{2\|\boldsymbol{x}\|y}{\sqrt{n_{\text{eff}}}}. \tag{32}$$

This minimum becomes unstable if the learning rate $\eta$ exceeds a critical value $\eta_c$, given by

$$\eta_c = \frac{\sqrt{n_{\text{eff}}}}{\|\boldsymbol{x}\|y} \tag{33}$$

It is worth noting that this is a necessary condition for $\lambda$ to oscillate around $2/\eta$. Otherwise, training converges to the zero-loss minimum with $\lambda = \lambda_{\min}$ for $\eta < \eta_{\max}$.

We can also derive the exact same result by analyzing the dynamics on the EoS manifold. As discussed in Section 5.2, the dynamics on the EoS manifold is given by the map $\Delta f_{t+1} = M(\Delta f_t)$, where

$$M(\Delta f) = \Delta f \left(1 - \frac{2\eta\|\boldsymbol{x}\|}{\sqrt{n_{\text{eff}}}}(\Delta f + y) + \left(\frac{\eta\|\boldsymbol{x}\|}{\sqrt{n_{\text{eff}}}}\right)^2 \Delta f(\Delta f + y)\right). \tag{34}$$

As demonstrated in Section 5.2, EoS in the UV model follows the period doubling route to chaos, with the period two cycle marking the onset. Hence, the conditions required for the emergence of the period two cycle are also necessary conditions for EoS. Consider the two-step dynamics on the EoS manifold given by the map $M^2(\Delta f) := M(M(\Delta f))$. This map has six fixed points (excluding three fixed points of the map $M$) summarized below

$$\Delta f^* = \frac{\eta\tilde{x}(1 - \eta\tilde{x}y) \pm \sqrt{\eta^2\tilde{x}^2(\eta\tilde{x}y - 1)(3 + \eta\tilde{x}y)}}{2\eta^2\tilde{x}^2} \tag{35}$$

$$\Delta f^* = \frac{3 + h(\eta,\tilde{x},y) \pm \eta\tilde{x}\left(\pm y + \sqrt{2}\sqrt{-\frac{-5+h(\eta,\tilde{x},y)+\eta\tilde{x}y(-2-\eta\tilde{x}y+h(\eta,\tilde{x},y))}{\eta^2\tilde{x}^2}}\right)}{4\eta\tilde{x}}. \tag{36}$$

Here $\tilde{x} = \frac{\|\boldsymbol{x}\|}{\sqrt{n_{\text{eff}}}}$ and $h(\eta,\tilde{x},y) = \sqrt{-7 + \tilde{x}y\eta(2 + \tilde{x}y\eta)}$. For the fixed points to exist, we require the expressions inside the square root to be non-negative, i.e.,

$$\left(\frac{\eta\|\boldsymbol{x}\|y}{\sqrt{n_{\text{eff}}}} - 1\right)\left(\frac{\eta\|\boldsymbol{x}\|y}{\sqrt{n_{\text{eff}}}} + 3\right) \geq 0 \implies \eta \geq \eta_1 = \frac{\sqrt{n_{\text{eff}}}}{\|\boldsymbol{x}\|y} \tag{37}$$

$$\frac{\eta\|\boldsymbol{x}\|y}{\sqrt{n_{\text{eff}}}}\left(\frac{\eta\|\boldsymbol{x}\|y}{\sqrt{n_{\text{eff}}}} + 2\right) - 7 \geq 0 \implies \eta \geq \eta_2 = \frac{(\sqrt{32} - 2)}{2}\frac{\sqrt{n_{\text{eff}}}}{\|\boldsymbol{x}\|y}. \tag{38}$$

As $\eta_1 < \eta_2$, the necessary condition for the period two cycle to emerge is $\eta > \eta_1 = \sqrt{n_{\text{eff}}}/\|\boldsymbol{x}\|y$, which coincides with the condition obtained earlier in this section.

## C.9 SUB-QUADRATIC LOSS

In this section, we detail how the UV model naturally leads to a sub-quadratic loss on the EoS manifold demonstrated by Ma et al. (2022).

First, recall that the EoS manifold that connects fixed points II and IV satisfies the relation $\lambda = 2\|\boldsymbol{x}\|(\Delta f + y)/\sqrt{n_{\text{eff}}}$. Then:

$$1.\ \text{Starting Relation:} \quad \lambda = \frac{2\|\boldsymbol{x}\|(\Delta f + y)}{\sqrt{n_{\text{eff}}}} \tag{39}$$

$$2.\ \text{Solving for } \Delta f: \quad \Delta f = \frac{\lambda\sqrt{n_{\text{eff}}}}{2\|\boldsymbol{x}\|} - y \tag{40}$$

$$3.\ \text{Factoring Out } y: \quad \Delta f = y\left(\frac{\lambda\sqrt{n_{\text{eff}}}}{2\|\boldsymbol{x}\|y} - 1\right) \tag{41}$$

$$4.\ \text{Defining } \eta_c: \quad \eta_c = \frac{\sqrt{n_{\text{eff}}}}{\|\boldsymbol{x}\|y} \tag{42}$$

$$5.\ \text{Expressing } \Delta f \text{ with } \eta_c: \quad \Delta f = y\left(\frac{\eta_c \lambda}{2} - 1\right) \tag{43}$$

$$6.\ \text{Computing the Loss:} \quad L = \frac{1}{2}\Delta f^2 = \frac{y^2}{2}\left(\frac{\eta_c \lambda}{2} - 1\right)^2 \tag{44}$$

As we mentioned in the main text, since $\lambda \sim O(\|\theta\|^2)$, the loss is sub-quadratic near its minimum.

## D  ADDITIONAL RESULTS ON SHARPNESS DYNAMICS

### D.1  SHARPNESS DYNAMICS OF NTP NETWORKS

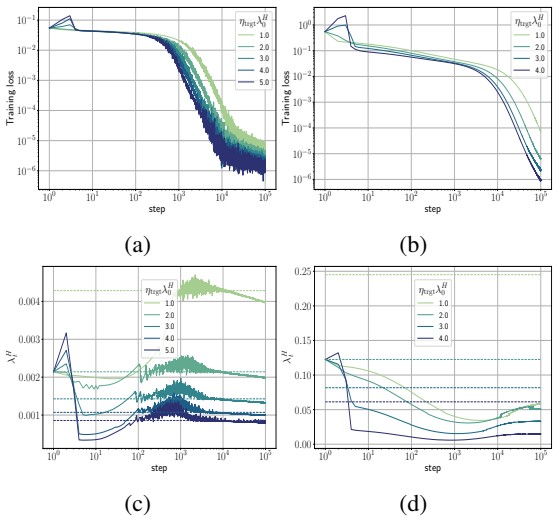

(a)

(b)

(c)

(d)

Figure 10: Training loss and sharpness trajectories of ReLU FCNs in NTP trained on a 5k subset of CIFAR-10 examples using MSE loss and GD: (a, c) $\sigma_w^2 = 0.5$, (b, d) $\sigma_w^2 = 2.0$.

Figure 10 shows that the sharpness dynamics of FCNs in NTP aligns with the behavior of FCNs in SP demonstrated in Figure 1.

### D.2  THE EFFECT OF BATCH SIZE ON THE FOUR TRAINING REGIMES

In this section, we examine the effect of batch size $B$ on the results presented in the main text. We find that our conclusions are robust for reasonable batch sizes around $B \approx 512$. For even smaller batch sizes, the dynamics becomes noise-dominated, and separating the inherent dynamics from noise becomes challenging. This observation further supports the use of SGD to reduce the computational cost of experiments in the subsequent sections involving CNNs and ResNets.

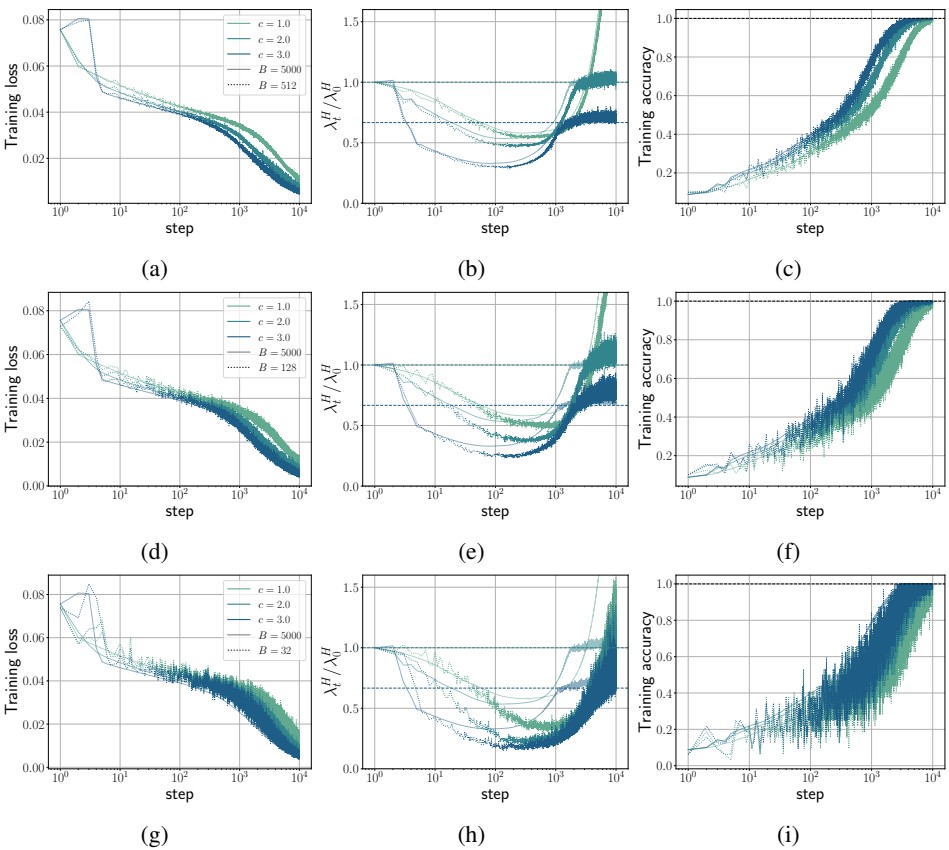

Figure 11: Comparison of SGD trajectories with their GD counterpart for a 3-layer FCNs in SP with $\sigma_w^2 = 0.5$ trained on a subset of CIFAR-10 consisting of $5,000$ training examples with MSE loss. The learning rate is scaled as $\eta = c/\lambda_0^H$ and batch sizes (a-c) $B = 512$, (d-f) $B = 128$, (g-i) $B = 32$ are considered. GD trajectories are plotted using solid lines with transparency.

Figure 11 shows that SGD trajectories of FCNs in SP begin to deviate from their GD counterpart significantly for batch sizes around $B \approx 128$. In contrast, for $\mu$P networks this deviation begins at a larger batch size of $B \approx 512$ as shown in Figure 12.

## D.3    GENERIZABILITY OF THE FOUR TRAINING REGIMES

This section shows that the four regimes are generically observed for different architectures, loss functions, and datasets. Figure 13 shows training trajectories of CNNs and ResNets trained SGD with batch size $B = 512$. Figure 14 show the training trajectories of Transformers trained on the WikiText-2 dataset using the cross-entropy loss. These results further exemplify that four regimes of training are generically observed for complex architectures, datasets, and loss functions.

## E    SHARPNESS-WEIGHT NORM CORRELATION

Section 5.1 reveals that several aspects of the training dynamics are controlled by the fact that, for a wide variety of initializations, at early times trajectories move closer to the saddle point II, resulting in an interim decrease in $\lambda$ (also proportional to the weight norm in this case), before eventually increasing. This critical point where all parameters are zero also exists in real-world models. We thus anticipate that in real-world models, the origin of the four training regimes may be related to a similar mechanism. This would predict a decrease in weight norm as training passes near the saddle point, followed by an eventual increase.

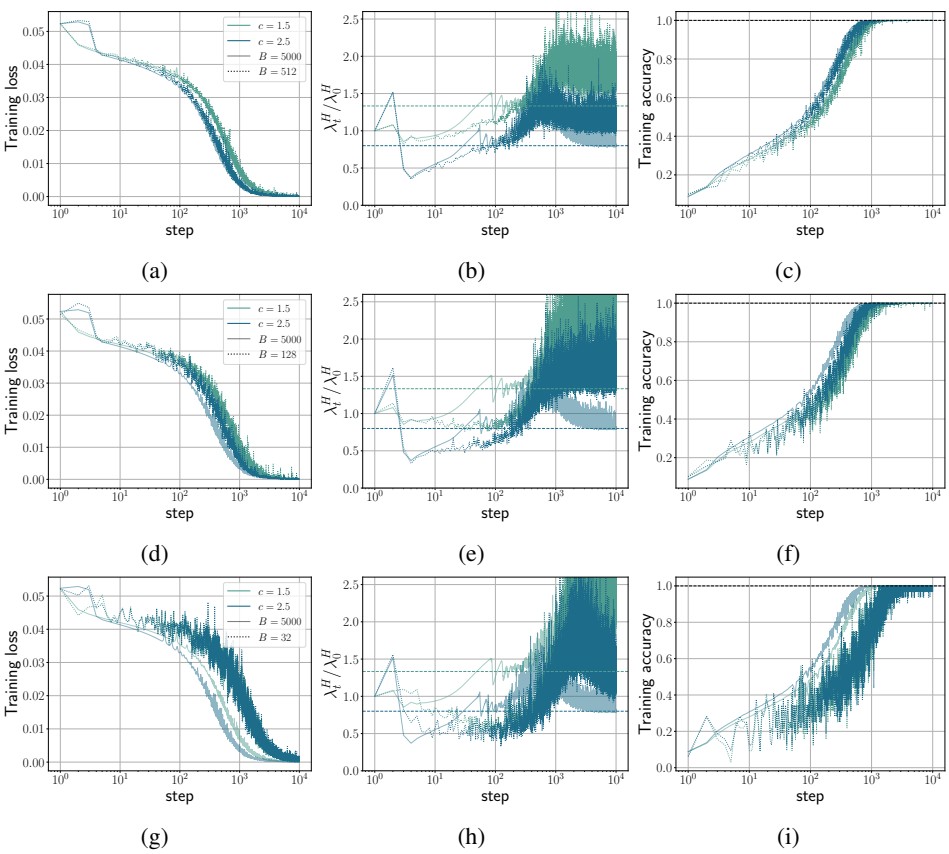

Figure 12: Same setting as Figure 11, except we used 3-layer FCNs in $\mu$P with $\sigma_w^2 = 2.0$.

Figure 15 validates this hypothesis. During the sharpness reduction and intermediate saturation regimes, we see a decrease in the weight norm, followed by an increase in the weight norm as the network undergoes progressive sharpening, following the prediction from the UV model. Similar correlations between the last layer weight norm and sharpness are utilized by Wang et al. (2022) to analyze the EoS phase. By comparison, we focus on the correlation between sharpness and weight norm during early training to attribute the emergence of four regimes to the critical point corresponding to all parameters being zero. In Appendix E, we provide further evidence for this correlation between sharpness and weight norm, extending this relationship to CNNs and ResNets.

This section presents additional results for Section 6, further supporting the relationship between sharpness and weight norm during training.

Figure 16 is an extended version of Figure 15, where we plotted the whole training trajectories and measured Pearson correlation

$$\text{Cor}(\|\theta_t\|, \lambda_t^H/\lambda_0^H) := \frac{\sum_{t'=1}^{t} \left( \theta_{t'} - \bar{\theta}_t \right) \left( \lambda_{t'}^H/\lambda_0^H - \overline{(\lambda^H/\lambda_0^H)}_t \right)}{\sqrt{\sum_{t'=1}^{t} \left( \theta_{t'} - \bar{\theta}_t \right)^2 \sum_{t'=1}^{t} \left( \lambda_{t'}^H/\lambda_0^H - \overline{(\lambda^H/\lambda_0^H)}_t \right)^2}} \tag{45}$$

Here $t \geq 2$ and $\bar{\theta}_t = (\sum_{t'=1}^{t} \theta_{t'})/t$.

Figure 17 shows the weight norm of each layer separately for the experiment in Figure 15. This result shows a high correlation between weight norm and sharpness through training.

We also confirm these correlations between weight norm and sharpness in CNNs for the experiment in Figure 13(a, b).

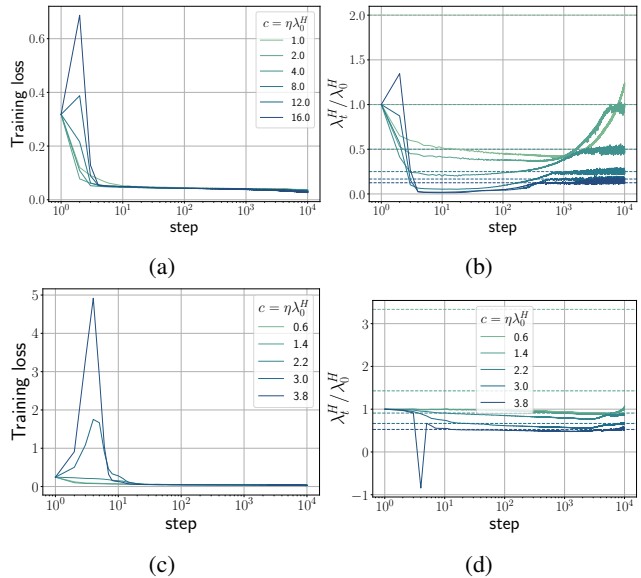

Figure 13: Training trajectories of (a, b) a 5-layer CNN in SP with $n = 64$, and (c, d) ResNet-18 with LayerNorm in SP, also with $n = 64$. Both models are trained on the CIFAR-10 dataset with MSE loss using SGD. The learning rate is scaled as $\eta = c/\lambda_0^H$ and batch size is $B = 512$. In panel (d), $\lambda_t^H$ becomes negative during early training. This is due to the power iteration method returning the largest eigenvalue by magnitude.

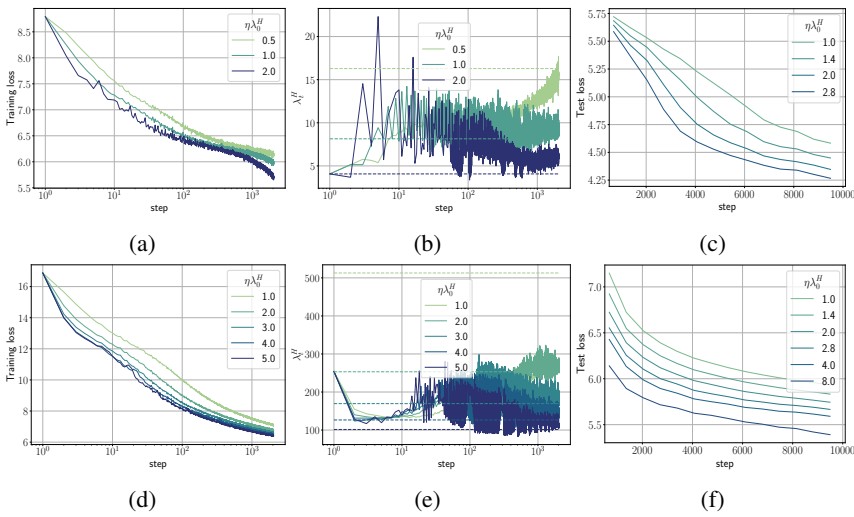

Figure 14: Training loss, sharpness and test loss trajectories of Transformers trained on Wikitext-2 using cross-entropy loss: (a, b, c) Pre-LN Transformers and (d, e, f) Pre-LN Transformer without last LayerNorm.

E.1 SETTING THE LAST LAYER TO ZERO ELIMINATES SHARPNESS REDUCTION

Figure 19 shows setting the last layer to zero eliminates sharpness reduction during early training.

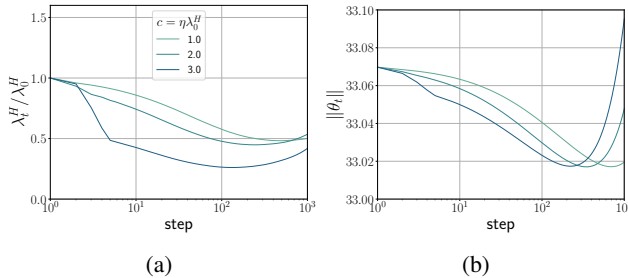

(a)          (b)

Figure 15: Sharpness and Weight Norm of 3-layer ReLU FCNs in SP with $\sigma_w^2 = 1/3$, trained on a subset of CIFAR-10 with $5,000$ examples using GD.

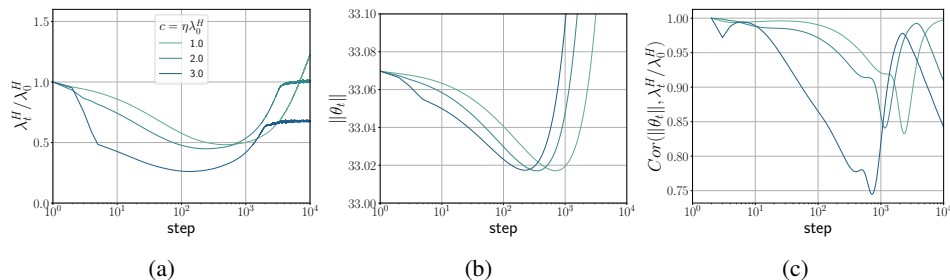

(a)          (b)          (c)

Figure 16: Sharpness and Weight Norm of 3-layer ReLU FCNs in SP with $\sigma_w^2 = 1/3$, trained on a subset of CIFAR-10 with $5,000$ examples using GD.

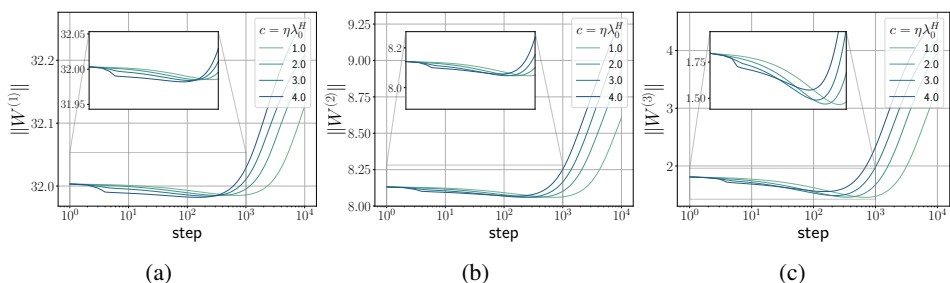

(a)          (b)          (c)

Figure 17: Weight Norm of each layer in 3-layer ReLU FCNs (same experiments as Figure 15): (a, b, c) SP with $\sigma_w^2 = 1/3$. All results are obtained by training on a subset of CIFAR-10 with $5,000$ examples using GD.

## F    ADDITIONAL PHASE DIAGRAMS OF EOS

This section demonstrates additional phase diagrams of EoS and quantifies the effect of batch size in the EoS regime. Figure 20 shows phase diagrams of EoS for FCNs trained on CIFAR-10 with MSE loss using SGD for $10,000$ steps for three different batch sizes. We observe that as the batch size decreases, $\lambda_t^H$ oscillates at a value different from $2/\eta$ depending on $\sigma_w^2$ and $s$. For large $\sigma_w^2$ and small $s$, $\lambda^H$ favors a smaller value for smaller batch size, which is in agreement with the observation in Cohen et al. (2021). In contrast, $\lambda^H$ can be larger than $2/\eta$ for small $\sigma_w^2$ and large $s$ at late training times.

Figures 21 and 22 show the phase diagrams of EoS for CNNs and ResNets trained on the CIFAR-10 dataset with MSE loss using SGD for $10,000$ steps with learning rate $\eta = c/\lambda_0^H$ and batch size $B = 512$. In contrast to the FCN phase diagrams, these architectures exhibit EoS behavior at smaller values of $s$ and larger values of $\sigma_w^2$, indicating their implicit bias towards EoS. Moreover, we observe in ResNets, that EoS is less sensitive to change $\sigma_w^2$, likely due to a combination of LayerNorm and residual connections Doshi et al. (2023).

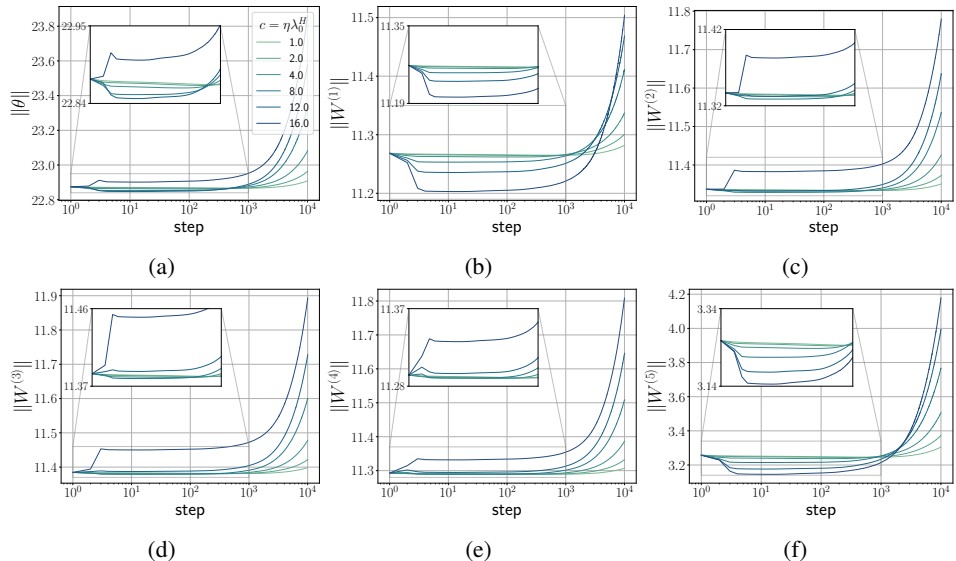

Figure 18: Weight Norm of each layer for 5-layer CNNs in SP (same experiments as Figure 13(a, b): (a) Total weight norm; (b-f) Weight norm of each layer. We see that for $c = 16$, the initial catapult in sharpness $\lambda^H$ (Figure 13(b)) is accompanied by a catapult in total weight norm. Notably, the total weight norm and per-layer weight norm, whether catapults (a, c-e) or not (b, f), show a decreasing trend during the early sharpness decreasing stage, followed by an eventual increase.

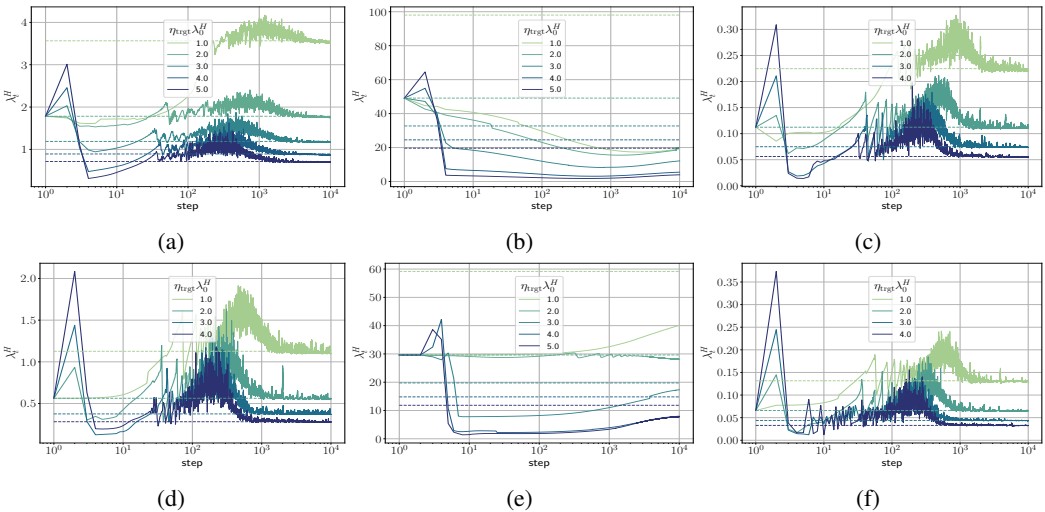

Figure 19: Sharpness trajectories of ReLU FCNs trained on a 5k subset of CIFAR-10 examples using MSE loss and GD. (a, b, c) SP with $\sigma_w^2 = 0.5$, SP with $\sigma_w^2 = 2.0$ and $\mu$P with $\sigma_w^2 = 2.0$. (d, e, f) the bottom row shows the effect of setting the last layer to zero at initialization. The dashed lines in the sharpness figures show the $2/\eta$ threshold.

It is worth noting that EoS boundaries in these phase diagrams are time-dependent. For instance, models close to the EoS boundary may eventually reach EoS on training longer (see Figure 13(b) $c = 1.0$ for example), causing a shift in the EoS boundary. Nevertheless, models with small learning rates, large $\sigma_w^2$, and small $s$ may never show EoS behavior, regardless of training duration, as predicted by the UV model and seen in Figure 1(e).

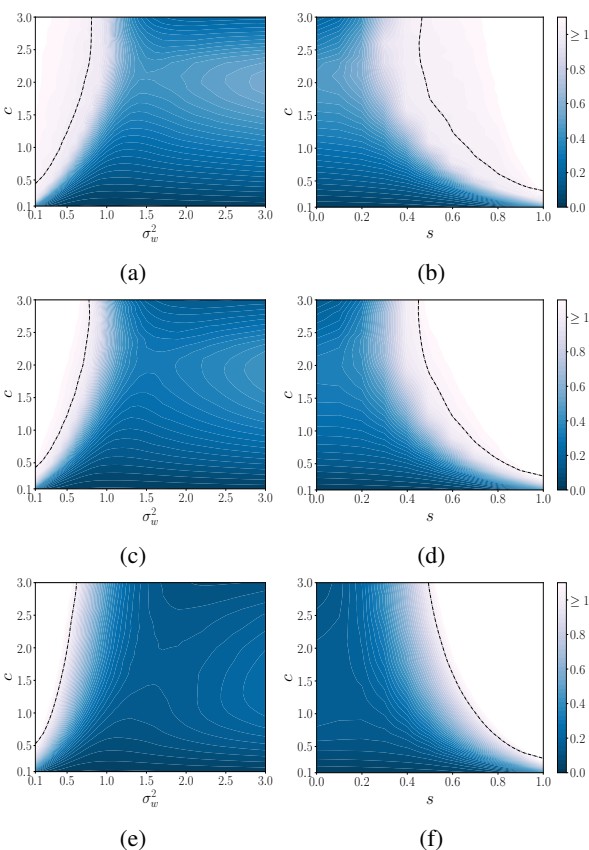

Figure 20: Phase diagram of EoS for 3-layer FCNs trained on CIFAR-10 with MSE loss using SGD with three different batch sizes: (a, b) $B = 512$, (c, d), $B = 128$, and (e, f) $B = 32$. The color indicates the value of $\eta \bar{\lambda}^H / 2$, where $\bar{\lambda}^H$ is obtained by averaging $\lambda_t^H$ over the last 200 steps. Except for the batch size, all settings are identical to Figure 4. Black dash-dotted lines indicate the phase boundary $\eta \bar{\lambda}^H / 2 = 1$. For clarity, these lines are generated from data smoothed with a Gaussian kernel.

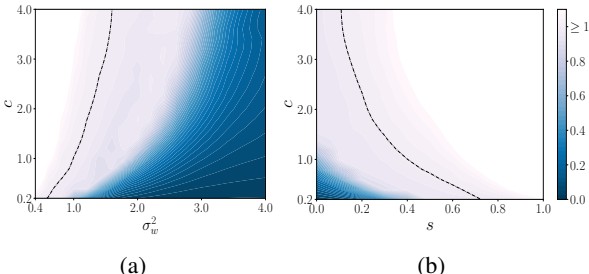

Figure 21: Phase diagram of EoS for 5-layer CNNs in SP with width $n = 64$ trained with MSE loss using SGD for $10,000$ steps with learning rate $\eta = c/\lambda_0^H$ and batch size $B = 512$.

# G  ROUTE TO CHAOS

## G.1  ROUTE TO EOS IN REAL DATASETS

This section presents additional bifurcation diagrams for different architectures and datasets. These results show the reminiscent of the period-doubling route to chaos observed in different architectures and datasets. In all figures, we choose the smallest and largest learning rate exhibiting EoS for

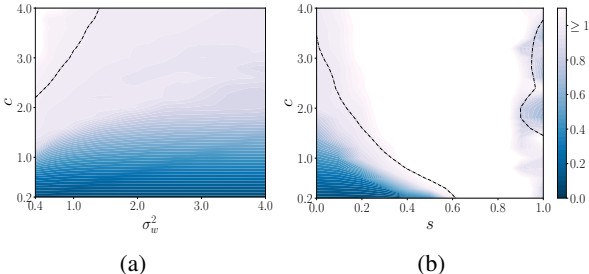

(a)                    (b)

Figure 22: Phase diagram of EoS for ResNet-18 in SP with width $n = 64$ trained with MSE loss using SGD for $10,000$ steps with learning rate $\eta = c/\lambda_0^H$ and batch size $B = 512$. For $s = 1$, the average eigenvalue $\bar{\lambda}^H$ is observed to be less than $2/\eta$. Upon detailed investigation of the trajectories, we found that $\lambda_t^H$ oscillates around a value lower than $2/\eta$. We leave this anomalous behavior as an observation.

plotting the trajectories and power spectrum. The structured route to chaos in realistic experiments can be disrupted due to a variety of reasons. Below, we discuss a few of them.

**Measurement of only the top eigenvalue of Hessian:** In our experiments, we only measured the top eigenvalue of the Hessian. However, when multiple eigenvalues of Hessian enter EoS, plotting only the top eigenvalue of Hessian is a projection that could obscure all the structured routes to chaos that the system may exhibit.

**The effect of correlations in real-world datasets:** Real-world datasets inherently contain correlations between different samples $(\boldsymbol{x}, \boldsymbol{y})$. These correlations can be quantified using the input-input covariance matrix $\Sigma_{XX} = XX^T \in \mathbb{R}^{d_{\text{in}} \times d_{\text{in}}}$ and output-input covariance matrices $\Sigma_{YX} = YX^T \in \mathbb{R}^{d_{\text{out}} \times d_{\text{in}}}$. In Appendix G.2, we find that a key determining factor in observing route-to-chaos is whether the power spectrum of $\Sigma_{XX}$ is flat or exhibits power law decay. We show that power-law decay in the singular values of the $\Sigma_{XX}$ results in long-range correlations in time and dense sharpness bands observed in real datasets.

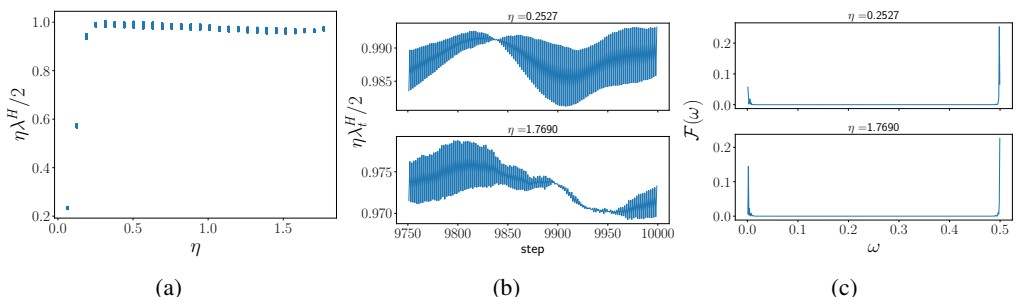

(a)                    (b)                    (c)

Figure 23: 4-layer FCN in SP with width $n = 512$ trained on a subset of 5000 examples of MNIST with MSE loss using GD. Both power spectrums are computed using the last 1000 steps of the corresponding trajectories.

## G.2    THE EFFECT OF POWER-LAW TRENDS IN DATA ON SHARPNESS TRAJECTORIES

In this section, we analyze a 2-layer linear FCN trained on the power law dataset described in Appendix B.1 to understand the origin of long-range correlations in sharpness trajectories and dense sharpness bands in realistic datasets.

Figure 28 shows the bifurcation diagram, late time trajectories, and the associated power spectrum of the network trained on the power-law dataset with the same $A_x = 1.0$ and $A_y = 1.0$, for four different combinations of power-law exponents: (i) $B_x = 0.0, B_y = 0.0$, (ii) $B_x = 1.0, B_y = 0.0$, (iii) $B_x = 0.0, B_y = 1.0$, and (iv) $B_x = 1.0, B_y = 1.0$. We observe that a power-law trend to the

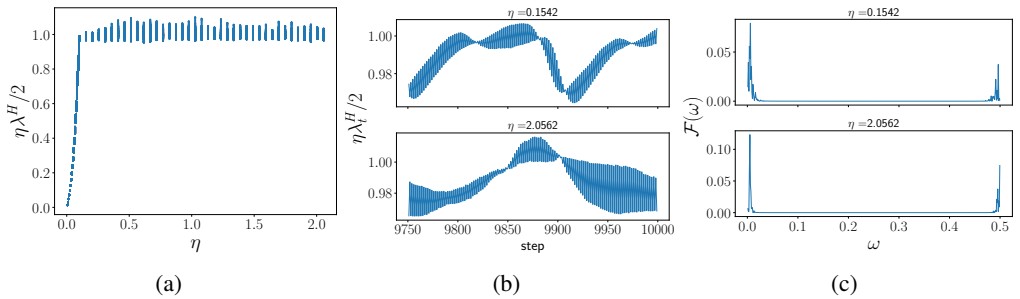

Figure 24: 4-layer FCN in SP with width $n = 512$ trained on a subset of $5000$ examples of Fashion-MNIST with MSE loss using GD. Both power spectrums are computed using the last $1000$ steps of the corresponding trajectories.

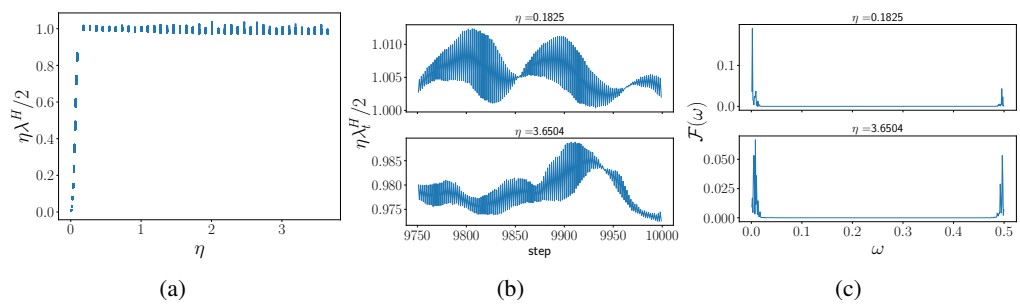

Figure 25: 4-layer FCN in SP with width $n = 512$ trained on a subset of $5000$ examples of CIFAR-10 with MSE loss using GD. Both power spectrums are computed using the last $1000$ steps of the corresponding trajectories.

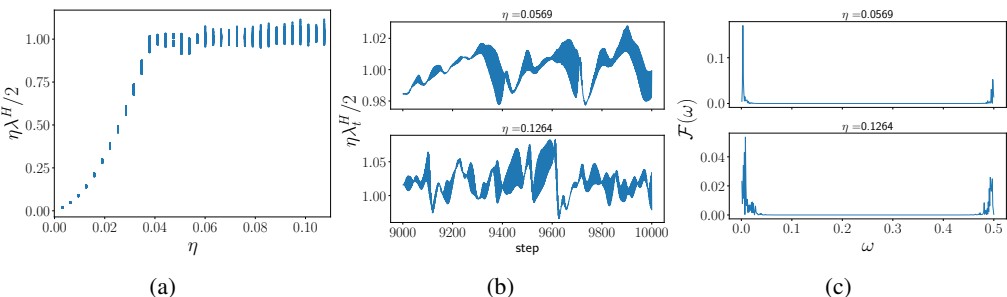

Figure 26: 5-layer CNN in SP with width $n = 32$ trained on a subset of $1000$ examples of CIFAR-10 with MSE loss using GD.

singular values of the input matrix results in dense sharpness bands observed in real datasets. It is worth noting that this is one way to obtain dense sharpness bands and in general, there can be many other methods.

## G.3 ROUTE TO CHAOS IN SYNTHETIC DATASETS

In this section, we analyze the route to chaos in synthetic datasets to gain insights into the dense sharpness bands in realistic datasets. We considered two datasets, defined as follows:

**Teacher-student dataset:** Consider a teacher FCN $f : \mathbb{R}^{d_{\text{in}}} \to \mathbb{R}^{d_{\text{out}}}$ with $d_{\text{in}} = 3072$, $d_{\text{out}} = 10$, depth $d$, and width $n = 512$ in Standard Parameterization. Then, we construct a teacher-student dataset $(X, Y)$ consisting of $P = 5000$ examples with $\boldsymbol{x}^{\mu} \sim \mathcal{N}(0, I)$ and $\boldsymbol{y}^{\mu} = \boldsymbol{f}(\boldsymbol{x}^{\mu}; \theta_0)$. Next, we train a student FCN with the same depth $d$ and depth $n$ as the teacher FCN on this dataset.

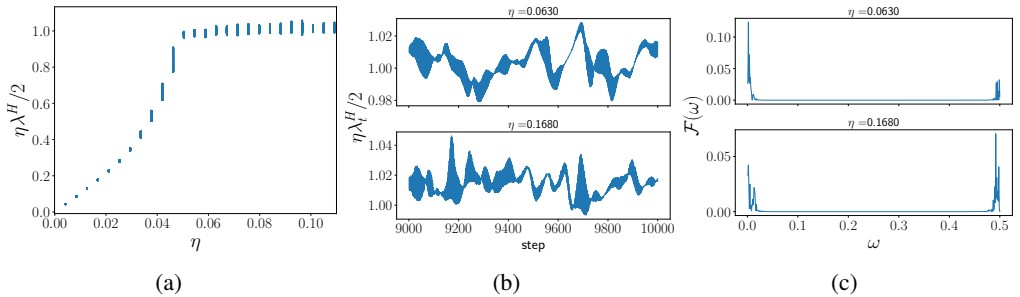

Figure 27: ResNet-18 in SP with width $n = 32$ trained on a subset of $1000$ examples of CIFAR-10 with MSE loss using GD.

Figures 29 and 30 show the bifurcation diagram, late time sharpness trajectories and the associated power spectrum of linear and ReLU FCNs trained on the teacher-student task. These figures show that while linear FCN shows the period-doubling route to chaos, ReLU FCN shows long-range correlations as observed in real datasets.

**Generative dataset:** Consider a $5$-layer CNN $\boldsymbol{f}(\boldsymbol{x}, \theta)$ in SP with $n = 64$, trained on the CIFAR-10 dataset with MSE loss using SGD with learning rate $\eta = {}^{12}/\lambda_0^H$ and momentum $m = 0.9$ for 100k steps. This model achieves a test accuracy of $76.9\%$. Then, we construct a generative image dataset $(X, Y)$ consisting of $P = 5000$ examples with $\boldsymbol{x}^\mu \sim \mathcal{N}(0, I)$ and $\boldsymbol{y}^\mu = f(\boldsymbol{x}^\mu; \theta)$. Next, we train an FCN in SP with depth $d$, width $n$, and weight variance $\sigma_w^2 = 0.5$ on the generated dataset.

Figures 31 and 32 show the bifurcation diagram, late time trajectories and the associated power spectrum of a $4$-layer ReLU FCN with linear and ReLU activations, trained on the generative CIFAR-10 dataset. We observe that while the linear network shows a period-doubling route to chaos, the ReLU shows long-range correlations as observed in real-datasets.

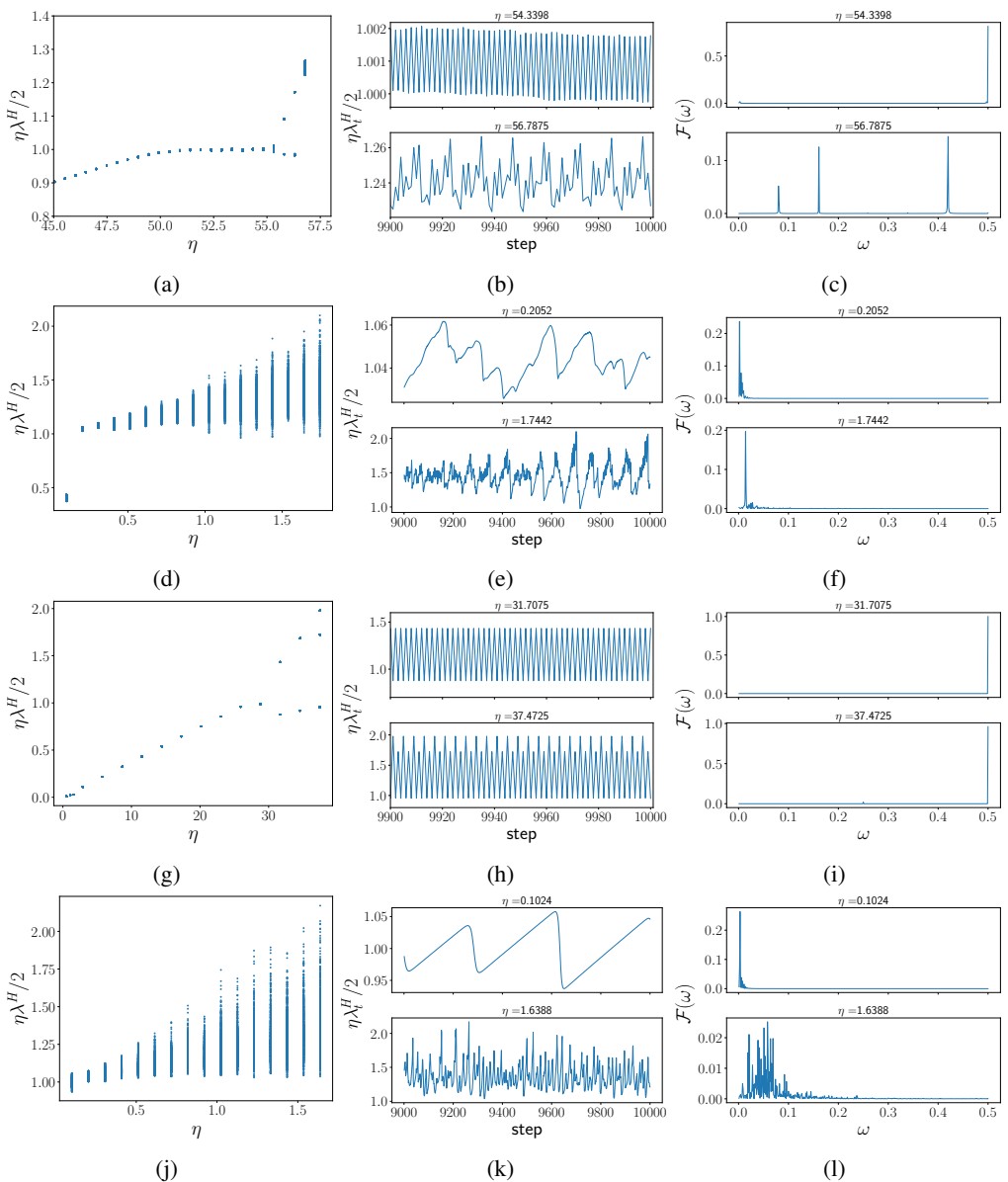

Figure 28: Bifurcation diagrams, late-time sharpness trajectories, and power spectrum of a 2-layer linear network trained on the power-law dataset for different parameter values: (a-c) $B_x = 0.0$, $B_y = 0.0$, (d-f) $B_x = 1.0$, $B_y = 0.0$, (g-i) $B_x = 0.0$, $B_y = 1.0$, and (j-l) $B_x = 1.0$, $B_y = 1.0$. All power spectrums are computed using the last 1000 steps of the corresponding trajectories.

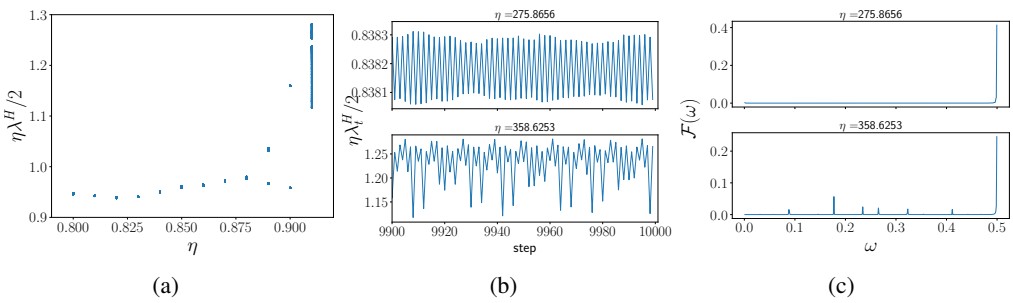

Figure 29: 2-layer linear FCN in $\mu$P trained on the teacher-student task. Both power spectrums are computed using the last 1000 steps of the corresponding trajectories.

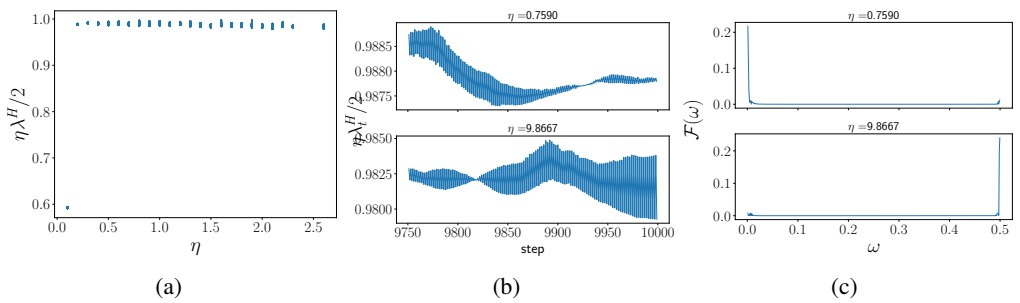

Figure 30: 4-layer ReLU FCNs in $\mu$P trained on the teacher-student task. Both power spectrums are computed using the last 1000 steps of the corresponding trajectories.

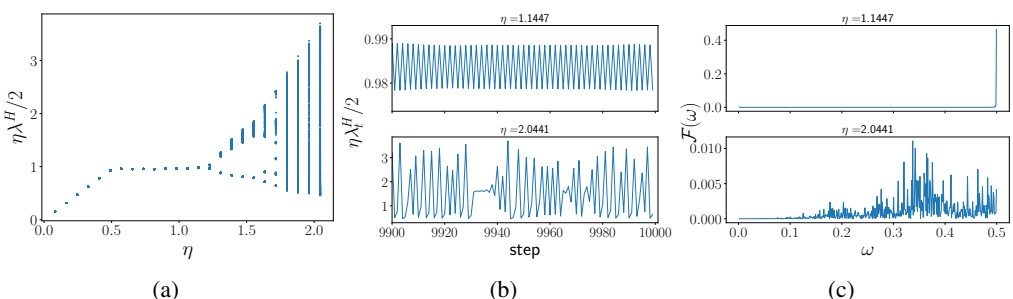

Figure 31: 4-layer linear FCNs in SP with $\sigma_w^2 = 0.5$ trained on the generative CIFAR-10 task with MSE loss using GD. Both power spectrums are computed using the last 1000 steps of the corresponding trajectories.

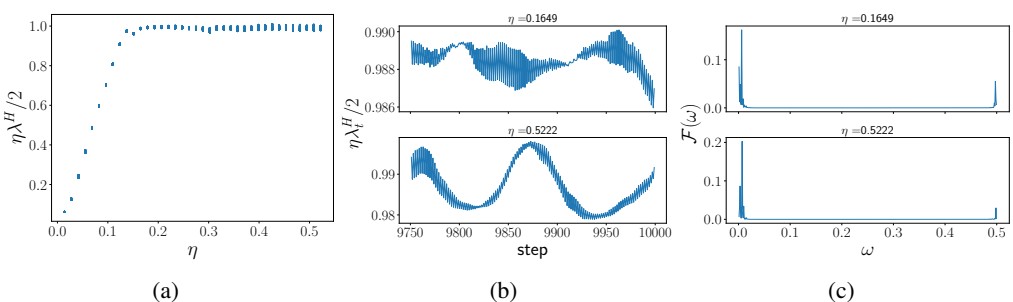

Figure 32: 4-layer ReLU FCNs in SP with $\sigma_w^2 = 0.5$ trained on the generative CIFAR-10 task with MSE loss using GD. Both power spectrums are computed using the last 1000 steps of the corresponding trajectories.

