# OpenReview forum: "Universal Sharpness Dynamics in Neural Network Training: Fixed Point Analysis, Edge of Stability, and Route to Chaos"
_ICLR.cc/2025/Conference — ICLR 2025 Poster_

### Official Review · Reviewer_GRmS · 2024-10-28

**Soundness:** 4
**Presentation:** 2
**Contribution:** 4
**Rating:** 8
**Confidence:** 2

**Summary:**

The paper analyzes four phases of training NNs (early time transient, intermediate saturation, progressive sharpening, edge of stability) from the perspective of the top eigenvalue of the Hessian of the loss. The authors show that these dynamics depend strongly on the parameterization of the network (width; standard vs. maximum parameterization) and the optimizer settings (large vs. small learning rate). This is not only done experimentally, but also in a linear toy model, the UV model. For this model, a fixed point analysis of the gradient dynamics can be performed, which shows that even this simple model can exhibit all four stages.

**Strengths:**

The paper is mathematically dense and, to the best of my knowledge, rigorous. The analysis of the edge of stability manifold as a chaotic attractor is indeed interesting, and so is the observation that EoS is correlated with improved performance. The analytical model for the UV system is valid and accessible. To the best of my (very) limited knowledge in this regard, this is the first analysis of this kind. Being not familiar with the related work, however, I am not very confident in my assessment.

The paper contributes to a better understanding of training neural networks by carefully studying a simple toy system and by showing that fully-connected NNs also exhibit similar properties. Via this connection, one can argue that the theoretical analysis indeed sheds light on the internal workings of NN training.

**Weaknesses:**

Among the weaknesses I would count (and please excuse my ignorance in this field):
- The analysis in Sec. 5.2 is quite simple and appears obvious or at least little surprising. While the observation that the EoS manifold is an attractor is quite relevant, the remaining statements (Result 1, Corollary 5.1, bifurcation diagrams) appear straightforward consequences of this observation. This is not a weakness of the paper in itself, but this observation could be used to shorten this section.
- It is not immediately clear how the observations made in this paper will affect deep learning engineering practices. In other words, some actionable insights would be appreciated, linking the insights from the paper to recommendations regarding learning rate, parameterization, and architectural choices.

**Questions:**

- What actionable insights can be derived from the analysis?
- Does Section 6 rely on the trace of the Hessian, or does it analyze its top eigenvalue? If it relies on the trace, can it be ensured that the qualitative picture is the same (i.e., do the results from Appendix C.5 for the UV model carry over to more complex architectures)?

---

> ### Author Response · Authors · 2024-11-19
>
> We thank the reviewer for their careful reading and encouraging comments. We will consider their suggestion to shorten Section 5.2 in the future version of the manuscript.
>
> >  What actionable insights can be derived from the analysis?
>
> Sharpness dynamics serves as a practical way to assess the initialization scale. When sharpness reduces during training, it implies that the model initialization was large in some sense. Instead, we could have picked a flatter initialization or performed a learning rate warmup to facilitate training at higher learning rates, which typically results in better performance. Additionally, the correlation between sharpness and weight norm dynamics offers a simpler way of assessing the initialization scale without the need to measure sharpness. If the weight norm decreases during training, it would suggest that the initialization was large.
>
> The UV model analysis suggests models initialized in the progressive sharpening region approach the zero-loss solution with low sharpness. This suggests using small initializations or parameterizations, which exhibit progressive sharpening from the start. Common practices such as setting the last layer to zero at initialization and using $\mu$P in the large-width limit regime result in zero model output at initialization, eliminating sharpness reduction during early training (see Figure 19 in the updated manuscript).
>
>
> > Does Section 6 rely on the trace of the Hessian, or does it analyze its top eigenvalue? ...
>
> In Section 6, we analyze the top eigenvalue of the Hessian. The trace of Hessian is only used in the UV model for mathematical simplifications. In Appendix C.5, we have verified that the trace of Hessian in the UV model is a good proxy for its sharpness.

---

> > ### Comment · Reviewer_GRmS · 2024-11-21
> >
> > I thank the authors for their response regarding acronyms information. As a non-expert in this specific field of research, I appreciate these comments. Given that my initial score was already quite high and my confidence is low, I will maintain my score.

---

### Official Review · Reviewer_yW4m · 2024-10-30

**Soundness:** 3
**Presentation:** 3
**Contribution:** 3
**Rating:** 6
**Confidence:** 3

**Summary:**

This work investigates training dynamics of a simple UV model and shows that it replicates several phenomenological properties that have been observed in larger networks. In particular, the same 4 "phases" of training are observed. The authors use the tractability of the UV model to identify why this occurs and when/why some of these phases can be "missed". The authors then show that some predictions from the UV model are found in larger models.

**Strengths:**

1. The paper is well written and motivated.

2. The numerical experiments are well designed and insightful.

3. The main conclusions of the analytical work are well explained and relatively easy to grasp.

4. The goal of understanding training dynamics is an important one that is well aligned with the NeurIPS community.

**Weaknesses:**

1. The use of simplified models is important and I appreciate that the authors demonstrated that the UV model, despite being very simple, has some of the same qualitative properties as what has been observed in other models. While the authors provide evidence that the features of their analysis can be used to understand training dynamics in larger models, I think there are a few weaknesses in these results:
    a. First, I think it is worth noting that (at least as far as I could see), comparing Fig. 1 and Fig. 7 (FCN vs. UV model), there seem to be some differences in behavior of the loss and $\lambda$. In particular, there are oscillations in the training loss that do not appear in the FCN, and there is less of a spike in the training loss and $\lambda$ for say the model with $\mu$P (Fig. 7C, F). This makes it unclear how "similar" the phenomenology are between the UV model and the training phases that have previously been studied. One way to try to quantify the "similarity" would be to use metrics for estimating topological conjugacy between training dynamics (arXiv:2302.09160).
    b. The authors show that $c$, $\sigma_W$, and $s$, which are predicted in the UV model analysis to play an important role in determining EoS are predictive of EoS in larger networks (e.g., Fig. 4). However, it seems that some parameters are much more important than others, e.g. for large enough value of $s$ any value of $c$ suffices to see EoS. This was not discussed in the main text and it was unclear to me why this would be. Similarly, the authors say there is a correlation between EoS and test accuracy, which is in my opinion weakly shown in Fig. 4. Maybe this is due to the fact that the color is saturated at 1? Or perhaps the way of estimating EoS ($\eta \lambda^H / 2$ ) is not the best way of estimating EoS?
     c. The lack of period doubling in large networks is a very interesting point (and the idea that the data could be suppressing it somehow is really cool!), but to me this suggests again the possibility that there are fundamental differences in the dynamics between the UV model (trained with one example) and modern architectures that makes it unclear how generally valid the analysis performed is. Providing more discussion on this would be helpful.

2. I did not follow how the authors got that the loss was $\mathcal{L}(\theta) = 1/2 \Delta f^2 = y^2/2 (\eta_c \lambda / 2 - 1)^2$ in Sec. 5 ("Connections to sub-quadratic loss"). Providing more detail would be useful.

3. The title of the paper "Universal sharpness" is perhaps too strong. I did not feel like the authors ever really made a claim of universality, nor did they provide much analysis of it, in the main text.

MINOR POINTS:
1. I was under the impression that the role of loss landscape sharpness and generalization was called into question by Dinh et al. (2017). Is there a reason this was not cited in the Introduction when discussing loss landscape and generalization?

2. In Appendix E, the examples used to validate the hypothesis about the weight norms is a 3-layer FCN, while Fig 5 has a 2-layer FCN. Is there a reason for these differences? Were the weight norms consistent with the hypothesis in 2-layer FCNs?

3. The authors mention that fixed point analysis is a generally powerful approach (Sec. 7). I think it would be good to discuss it's limitations more, particularly for nonlinear DNNs.

4. Very minor, but I think putting references in parantheses would make the paper a little easier to read. I.e., "the early training dynamics [Kalra and Barkeshli (2023)]" (page 1).

**Questions:**

Q1. How truly "similar" are the dynamics in the UV model and the FCNs used to motivate this paper?

Q2. Does the analysis performed by the authors demonstrate "Universal" properties?

---

> ### Author Response · Authors · 2024-11-19
>
> We thank the reviewer for the detailed comments and highly constructive suggestions. In response to the reviewer's comments, we have included a thorough discussion on the limitations of the UV model in Section 7. Below, we address these concerns individually.
>
>
> > ...comparing Fig. 1 and Fig. 7 (FCN vs. UV model), there seem to be some differences in behavior of the loss and $\lambda$...How truly "similar" are the dynamics in the UV model and the FCNs used to motivate this paper?...
>
> As correctly noted by the reviewer, the UV model does not capture two aspects of the training dynamics.
>
> First, it does not capture the non-monotonic decrease in sharpness at EoS. This is because the dynamics is fully characterized by two variables and the EoS condition $\lambda \sim 2/\eta$ puts one more constraint on the dynamics. Resultantly, the dynamics is described by only a single variable, and loss and $\lambda$ oscillates at EoS. This is in contrast to real-world models where the training converges along the stable directions at EoS. Therefore, the UV model should be thought of as a minimal model of the top eigenspace oscillations at EoS.
>
> Second, the loss may not catapult for the $\mu$P case shown in Figure 7(c, f) as pointed out by the reviewer. This arises because the initialization $\lambda_0 = 2$ is comparable to the maximum trainable learning rate $\eta_{\text{max}} = 1.0$ (for loss increase at initialization, we require $\eta > 2 / \lambda_0 = 1$). Such situations are also observed in practice. Such as Pre-LN Transformers trained with SGD, as shown in Figure 14(a, c) of the paper.
>
>
> > ..it seems that some parameters are much more important than others, e.g. for large enough value of $s$ any value of $c$ suffices to see EoS...
>
> As pointed out by the reviewer, parameters such as $s$ are more important in observing EoS and the UV model does not make any such predictions. We believe this is due to the non-linear relationship between the initial sharpness and $\sigma^2_w$, $s$ and $c$. While the UV model does not capture these complex relationships, it does predict that these parameters are crucial in determining EoS.
>
> The weak correlation between performance and EoS arises from clipping $\eta \lambda^H_t / 2$ at $1$. We have updated Figure 4 to show $\eta \lambda^H / 2$ without any cap. These results show a better correlation between test accuracy and EoS.
>
>
> > The lack of period doubling in large networks is a very interesting point....
>
> As observed by the reviewer, Section 6.4 demonstrates that real-world data suppresses the period doubling route to chaos. Nevertheless, we do observe a reminiscent of period doubling route to chaos in the power spectrum of sharpness trajectories of real-datasets (peaks in power spectrum Figure 5(f) at $\omega=1/2, 1/4, 1/8$).
>
> This is a limitation of the UV model, which is trained on a single example and thus cannot capture the complex correlations between training examples. We believe highlighting this limitation and identifying its root cause is important. Nevertheless, the UV model successfully captures various other features of sharpness dynamics, such as progressive sharpening, early sharpness reduction, and late-time sharpness oscillations around $2/\eta$. The emergence of these phenomena in the simplified UV model suggests they are fundamental properties of neural network training dynamics.
>
> > I did not follow how the authors got that the loss was ...
>
> We use the relation that on EoS manifold $\lambda = 2 \|x \| (\Delta f + y) / \sqrt{n_{\mathrm{eff}}}$, to substitute $\Delta f$ by $\lambda$ and other variables, then we pull target $y$ out of the parentheses and combine other variables into $\eta_c = \sqrt{n_{\mathrm{eff}}} / \|x\|y$. We have updated section C.9 for detailed derivations.
>
> > The title of the paper "Universal sharpness" is perhaps too strong...Does the analysis performed by the authors demonstrate "Universal" properties?..
>
> We chose `Universal sharpness' as the title because several prior works were operating in different sharpness regimes. For instance, (Lewkowyz et al. 2020, Zhu et al. 2022) operated in the sharpness reduction regime, whereas (Jastrzebski et al. 2020, Cohen et al. 2022) operated in the progressive sharpening regime. Our work resolves this discrepancy by providing a unifying analysis for different sharpness regimes, motivating the chosen title. Furthermore, these sharpness trends are observed across architectures, including small CNNs, ResNets, and Transformers, where sharpness computation is feasible.

---

> ### Author Response · Authors · 2024-11-19
>
> > I was under the impression that the role of loss landscape sharpness and generalization was called into question by Dinh et al. (2017)....
>
> We thank the reviewer for pointing out this important reference. We have added it to the first paragraph of the introduction.
>
> > In Appendix E, the examples used to validate the hypothesis about the weight norms is a 3-layer FCN, while...
>
> Yes, the weight norms were consistent with the two-layer FCNs results. We analyzed the weight norm of models with varying depths to verify that the correlation was not limited to two-layer networks.
>
> > The authors mention that fixed point analysis is a generally powerful approach ..
>
> As we mentioned in Sec. 7, the limitation of applying this method is the closure of the dynamical equations describing the model. For deep linear networks, this model can be generalized by introducing more function space variables. For non-linear DNNs, the function space dynamics are hard to close with a few variables. We have included this discussion in Section 7.
>
> > I think putting references in parentheses would make the paper a little easier to read..
>
> We thank the reviewer for their suggestion, which we will incorporate in the future version of the manuscript.

---

> > ### Comment · Reviewer_yW4m · 2024-11-21
> >
> > I thank the authors for their detailed response. I appreciate the authors pointing out the results of the Pre-LN Transformer (Fig. 14) and confirming that their results held for a range of FCN depths. In addition, with the authors removing the cap in Fig. 4, I am much more convinced of the correlation the authors claimed.
> >
> > For these reasons, I am willing to increase my score. However, some of my concerns on the extent to which minimal model captures aspects of the training dynamics in larger scale networks remains. If the authors have any additional insight or can provide another experiment that demonstrates the utility of the UV model, that would greatly help.

---

### Official Review · Reviewer_AyGJ · 2024-11-03

**Soundness:** 3
**Presentation:** 1
**Contribution:** 2
**Rating:** 5
**Confidence:** 3

**Summary:**

This paper covers a tremendous amount of theoretical work, presenting a simple model which the authors analyze to study the role of initialization and parameterization on initial movements in sharpness during training; the role of initialization, parameterization, and learning rate on the existence of the Edge of Stability; and finally drawing analogy between the EoS and period-doubling bifurcations.

**Strengths:**

The Edge of Stability (EoS) is an important phenomenon observed in recent research, spurring questions about the understanding of dynamics for non-convex optimisation with deep neural networks. This paper studies the early dynamics of these networks, providing insights on the early movements in sharpness and the existence of EoS later on in the training trajectory.

The theoretical analysis is clear, and the resulting phase diagram in Section 4 appears novel, and is insightful and easy to understand.

The applicability of the analysis appears general to a number of common parameterizations and can be used to draw insights to deep neural networks in general.

**Weaknesses:**

While this work presents impressive analysis, revealing insights toward important phenomena in optimization, the empirical support for individual claims are less apparent, possibly as a combination of limited empirical evidence and poor presentation. Additionally, the theoretical approach is hugely similar to existing work (Lewkowyz, 2020), though many additional insights revealed, especially concerning early dynamics. Finally, the connection of this work to common motivations for optimization analysis (e.g. generalization, feature learning) is indirect - only the suggestion that the existence of EoS can improve generalization error is presented in Fig 4 and discussed sparingly.

The areas for improvement in empirical support are detailed as follows:
1. Section 6.1 claims that $\mu P$ networks begin training in a flat region of the landscape where gradients point to increased sharpness, while NTP/SP networks experience a reduction, the evidence in Fig. 1 suggests that this distinction is not so clear.
2. Section 6.2 suggests that real-world deep neural networks (sometimes) undergo an interim decrease in $\lambda$ and the weight norm, but the real-world models presented in Fig. 1 and App. E exhibit a decrease in weight-norm. As real-world initializations typically use 0-mean weight initializations, one might expect that at least some of the models can be initialized in the R1 regime and immediately enter EoS. Therefore, the evidence that initialization in R1 can exist for real-world networks is limited, questioning the extent to which the theoretical model can apply to real-world models.
3. In general, the oscillation of EoS is normally observed around $2/\eta$, which is also stated in the paper in Sections e.g. 1 and 3. However, the point of oscillation predicted by the UV model (as seen in the phase diagrams) do not necessary have to be near 2/\eta. This distinction does not appear to be sufficiently addressed.
4. Some of the empirical evidence discusses the effects of SGD, but it is an open question how and whether EoS phenomena translates to SGD. The methodology at which EoS is computed for SGD experiments in the Appendix are extremely unclear. An important scaling for the learning rate as $\eta = c/\lambda_0$ is stated, but not justified or explained.

**Questions:**

1) Under light investigation, the mathematical derivations for Eqs. 1 and 2 are sound. However, one might expect that the dynamics of the theoretical model might be independent to the sign of $\delta f_t$. Can you provide an intuition to why this might not be the case?
2) What is the justification for setting $\eta = c/\lambda_0$?
3) What is a 'low/minimal sharpness bias', as in Section 6.1?
4) What can I do to load this PDF more quickly?

**Details Of Ethics Concerns:**

This is a theoretical paper, no ethics review is needed.

---

> ### Author Response · Authors · 2024-11-19
>
> We thank the reviewer for their detailed comments.
>
> > ..the theoretical approach is hugely similar to existing work (Lewkowyz, 2020), though many additional insights revealed
>
> While we use a similar model compared to Lewkowycz et al. (2020), our analysis differs in two key aspects:
>
> 1. Our work recognizes the crucial role of non-zero target $y$ and parameterization. Lewkowycz et al. (2020) set $y = 0$ (Eqs. 6-7 of their paper), which results in a model that cannot exhibit progressive sharpening (and consequently EoS) for any initialization or parameterization. In comparison, we observe progressive sharpening and EoS, precisely because of non-zero target $y$ and $\mu$P.
>
> 2. Furthermore, their analysis relies on specific approximations to simplify the dynamical equations to analyze catapult dynamics, particularly leveraging the existence of $\mathcal{O}(1/n)$ terms that can be ignored in the NTK regime. This approach cannot be generalized to $\mu$P ($n_{\text{eff}} = 1$) where such approximations are infeasible. In contrast, our generalized UV model dynamics comprehensively captures different sharpness phenomena across parameterizations.
>
> > Section 6.1 claims that $\mu$P networks begin training in a flat region of the landscape where gradients point to increased sharpness...
>
> We thank the reviewer for highlighting this potential source of confusion.
>
> For the same variance ($\sigma^2_w = 2.0$), the two parameterizations show completely different dynamics. Excluding large learning rates resulting in loss and sharpness catapults, models in NTP/SP experience sharpness reduction, whereas $\mu$P exhibits progressive sharpening after one step of reduction. From the UV model, if we set $f_0 = 0$, then the model is necessarily initialized in the progressive sharpening regime, which is satisfied by $\mu$P at large width. However, due to finite width corrections, training can be initialized in the sharpness reduction regime and may experience a slight sharpness reduction. To validate this, we set $f_0 = 0$ for models in Figure 1, by setting the last layer to zero at initialization. Figure 19 shows that setting the last layer to zero completely eliminates sharpness reduction in all cases. This result suggests that the initial one-step sharpness reduction in $\mu$P networks observed in Figure 1(f) is likely due to finite width effects.
>
> We note that this distinction extends beyond the choice between NTP/SP and $\mu$P; the initialization scale $\sigma_w$ also influences the initial conditions, potentially leading to different sharpness trajectories. In Figure 1 (a, d), we deliberately chose a small enough $\sigma_w$ value with SP that exhibits all four training regimes, including the initial sharpness reduction. If we consider an even smaller value of $\sigma_w$, then training exhibits an even smaller sharpness reduction, and the dynamics aligns more with the $\mu$P case.
>
> These outcomes align with our UV model's predictions.
>
> > Section 6.2 suggests that real-world deep neural networks (sometimes) undergo an interim decrease in $\lambda$...
>
> We request the reviewer to clarify the following concern:  ``Section 6.2 suggests that real-world deep neural networks (sometimes) undergo an interim decrease in $\lambda$ and the weight norm, but the real-world models presented in Fig. 1 and App. E exhibit a decrease in weight-norm.''
>
> Regarding initialization in regime R1, can the reviewer also clarify what they imply by " initialization in R1 can exist for real-world networks is limited,"?
>
> While we await your clarification, we would like to take the opportunity to clarify the initialization in the R1 region for NTP/SP and $\mu$P models in a real-world setting. NTP/SP models are unlikely to be initialized in the R1 regime in real-world scenarios because in high dimensions the initialized function will be concentrated around the variance for a zero-mean initialization with high probability. In comparison, the initial variance of the $\mu$P models scales as $1/n$, and for sufficiently large widths, they can be initialized in the R1 regime. The UV model does not capture these high-dimensional cases which is a limitation.

---

> > ### Author Response · Authors · 2024-11-19
> >
> > > In general, the oscillation of EoS is normally observed around $2/\eta$, ....
> >
> > In the UV model, the oscillations do occur around $2/\eta$ as seen in Figure 3, consistent with real-world observations. The key difference is the magnitude of these oscillations around $2/\eta$, with the UV model exhibiting higher oscillations than we observe in real-world models. As shown in Figure 5(b), real-world models also exhibit increased fluctuations with learning rates. We believe that training diverges before exhibiting such extreme fluctuations, similar to the numerically observed bifurcation diagram of the UV model shown in Figure 3(b).
> >
> > > Some of the empirical evidence discusses the effects of SGD...
> >
> > While we agree with the reviewer that the effects of SGD on the stability threshold are not well understood, our Appendix experiments are restricted to batch sizes that closely track the GD counterparts. In the Appendix, we performed two sets of experiments with SGD. First, we show that the existence of four regimes is robust up to reasonable batch sizes $B \approx 512$. In such cases, we observe that the SGD trajectories closely track the GD ones, as discussed in Appendix D.2. This motivated the use of SGD to reduce the computational cost of the EoS phase diagrams involving CNNs and ResNets. Figure 13 shows that the EoS threshold for CNNs and ResNets trained with batch size $B = 512$ is very close to $2/\eta$. Therefore, $\eta \lambda^H_t / 2$ is a reasonable proxy for the EoS phase diagrams shown in Figures 20 and 21.
> >
> > After careful investigation, we found a typo regarding the route to chaos experiments. The route to chaos experiments for CNNs and ResNets are performed with GD, as mentioned in the respective figures, and not with SGD, as mentioned at the beginning of Appendix G.1. We thank the reviewer for their question and we have fixed this typo in the updated manuscript.
> >
> > > Under light investigation, the mathematical derivations for Eqs. 1 and 2 are sound....
> >
> > Contrary to intuition, the non-zero target $y$ breaks the degeneracy between the zero loss fixed points and the saddle point II corresponding to zero parameters. This results in a fixed point structure and dynamics that are not symmetric around $\Delta f$, which is crucial for observing progressive sharpening and EoS in this model.
> >
> > > What is the justification for setting $\eta = c / \lambda_0$?
> >
> > The initial sharpness heavily varies, up to two orders of magnitude, depending on the initial weight variance $\sigma^2_w$ and parameterization. Due to this variation, a small learning rate for one parameterization can result in divergence for another. The Descent Lemma (Damian et al. 2022) reveals that $\eta \lambda^H$ has no units, and suggests sampling $c = \eta \lambda^H$ as a systematic way of selecting learning rates across parameterizations. For constant learning rate schedules, using the initial sharpness $\lambda_0$ as the normalization factor emerges as a natural choice.
> >
> > > What is a ’low/minimal sharpness bias’, as in Section 6.1?
> >
> > By low/minimal sharpness bias, we mean that for $\mu$P training starts with a small initial sharpness $\lambda < 2 / \eta_c$. Resultantly, the sharpness increases, and training converges to nearby zero-loss solutions with small sharpness ($\lambda \approx 2/ \eta_c$). In comparison, for NTP training begins at relatively high sharpness and may converge to nearby zero loss solutions with sharpness $\lambda > > 2 / \eta_c$.
> >
> > > What can I do to load this PDF more quickly?
> >
> > We found that opening the PDF in a lightweight PDF reader helps.

---

> > > ### Author Response · Authors · 2024-11-25
> > >
> > > We thank the reviewer for their thoughtful review and constructive comments. We would appreciate your thoughts on whether our responses have addressed your questions and remain available for clarifications if needed.

---

> > > > ### Comment · Reviewer_AyGJ · 2024-11-25
> > > >
> > > > I thank the authors for addressing my review in a careful manner and am grateful that they have clarified some of my concerns. I remain impressed by the level of analysis and work in this paper, but the primary limiting factors remains the impact from this work, with its similarities with previous work and limited connection to common motivations for studies in this area are not sufficiently addressed. As a result, I am going to maintain my score.
> > > >
> > > > I am grateful for the authors pointing out the confusion in my comment with regard to initializations in R1. To clarify, it isn't clear to me that real-world networks are initialized in the R1 regime, which is shown to have an immediate increase in $\lambda$ and weight norm. The authors show evidence for initializations in the R2 regime, where $\lambda$ and weight norm decrease. It seems like real-world models are initialized from zero-mean distributions, which is close to Saddle Point II when naively translated to the theoretical model. However, there is limited evidence to show that real-world models do initialize in this space and undergo the trajectories described by the theoretical analysis in the paper, which may be a limitation for this model.

---

> > > > > ### Author Response · Authors · 2024-11-26
> > > > >
> > > > > We thank the reviewer for the clarifications and further comments.
> > > > >
> > > > > > ..it isn't clear to me that real-world networks are initialized in the R1, which is shown to have an immediate increase in $\lambda$
> > > > >
> > > > > Real-world models can be easily initialized in the R1 region. Below, we detail two distinct real-world settings that exhibit progressive sharpening right from initialization.
> > > > >
> > > > >
> > > > > First, Figure 19 of the updated paper shows that real-world models can be easily initialized in the R1 region by setting the last layer to zero at initialization, which is a commonly used heuristic in practice[1]. Specifically, we consider the experimental setup in Figure 1 and set the last layer to zero at initialization. These models exhibit progressive sharpening right from the start, completely eliminating initial sharpness reduction in all three cases (including SP with $\sigma^2_w = 0.5$ and $\mu$P).
> > > > >
> > > > > We can also understand this using the UV model. On setting the last layer to zero at initialization, we get $f_0 = 0$, which results in the following update equation for $\lambda$:
> > > > >
> > > > > $$ \lambda_{1} = \lambda_0 \left(1+ \frac{\eta^2 y^2 \||\bf{x}\||^2}{n_{\text{eff}}} \right),$$
> > > > >
> > > > >
> > > > > Next, updated Figure 14 shows that the commonly used Pre-LN Transformers exhibit progressive sharpening right from the start and enter EoS. Meanwhile, if we remove the last LayerNorm, the initial sharpness is $100$ times higher and exhibits sharpness reduction during early training. This suggests that commonly used Pre-LN Transformers have a smaller sharpness at initialization and exhibit progressive sharpening at initialization.
> > > > >
> > > > > These results demonstrate that real-world models can be easily initialized in the R1 regime.
> > > > >
> > > > > [1] https://github.com/KellerJordan/modded-nanogpt/blob/master/train_gpt2.py#L258C14-L258C21
> > > > >
> > > > > > ...limited connection to common motivations...
> > > > >
> > > > > To address the reviewer's concerns regarding connections to generalization, we expanded Figure 14 to include two additional panels (c, f) that show the test performance of Transformers trained on WikiText-2 for $10,000$ training steps. These results demonstrate that Pre-LN Transformers without the final LayerNorm (which exhibit sharpness reduction during early training) achieve significantly higher test losses as compared to the commonly used Pre-LN Transformers with the final LayerNorm. These extended experiments provide good empirical support for the correlation between sharpness values and generalization.
> > > > >
> > > > > Regarding the reviewers' concern about feature learning: By changing the effective width $n_{\text{eff}}$ in the UV model, we revealed that $\mu$P with $n_{\text{eff}}=1$, exhibits a more drastic change in sharpness. In the UV model, this means the internal feature gets updates, which translates to feature learning in real-world scenarios. Figure 4(d) shows that FCNs with a smaller effective width (large $s$) attain better performance.
> > > > >
> > > > > > .. its similarities with previous work..
> > > > >
> > > > > We would like to take the opportunity to further clarify the differences between our work relative to Lewkowyz et al. 2020.
> > > > >
> > > > > Our theoretical analysis of the function space equations is **completely different** from that of Lewkowyz et al. 2020. Specifically, we employ fixed point analysis, which enables us to analyze the **complete** dynamics without any approximations.
> > > > > Furthermore, Lewkowyz et al. 2020 restrict their analysis to catapult dynamics. In comparison, we analyze various other phenomena including sharpness reduction, progressive sharpening, and EoS.
> > > > > Therefore, the theoretical analysis and the phenomena analyzed are significantly different from the referenced study.

---

### Official Review · Reviewer_Knqc · 2024-11-05

**Soundness:** 3
**Presentation:** 3
**Contribution:** 3
**Rating:** 6
**Confidence:** 3

**Summary:**

The paper studies nonlinear gradient descent dynamics in a two-layer linear model trained on a single example. The model in question (a "UV" model) extends the model analyzed in Lewkowycz et al. (2020) by including a nonzero target scalar and incorporating different parameterizations (standard / neural tangent / $\mu P$) via an effective width parameter. The resulting gradient descent dynamics has a rich phase portrait with several stable/unstable fixed points/lines and different evolution patterns depending on initial conditions and hyperparameters. In particular, the authors describe the absence or presence of a catapult regime, sharpness reduction, progressive sharpening, edge of stability oscillations. For the edge of stability (EoS) regime, it is shown to occur on a particular line in the parameter space at learning rates above a particular critical value, and its origin is related to the well-known period-doubling fractal mechanism. After that, the paper discusses a number of predictions one can make for realistic models based on the obtained phase picture of the toy model: a correlation of the parameterization with sharpness, temporary decrease of the weight norm, conditions of EoS and sharpness fluctuations.

**Strengths:**

The paper is easily readable and clear. A substantial background and literature review are provided. A large number of appendices provide substantial additional details of the methodology, computations, and experiments.

The toy model proposed in the paper is simple, but sufficiently rich to describe multiple complex effects observed in realistic models (catapult regime, sharpness reduction, progressive sharpening, edge of stability oscillations). The paper gives a systematic discussion of how these effects depend on the initialization, parameterization, learning rate.

The paper uses the toy model to conjecture several qualitative predictions on the learning behavior in realistic models/data. These are illustrated by multiple experiments with different networks and data.

**Weaknesses:**

I generally like the paper, but I'm not convinced in the sufficient novelty and significance of its results. The UV model studied in the paper is a toy model and is only a slight generalization of the well-known model from Lewkowycz et al. (2020). The study of this model is a fairly straightforward analysis of the 2D phase portrait of a simple $(\Delta f, \lambda)$ dynamics with respect to several hyperparameters. The paper does not develop any new analytic methods or prove any new general theorems. All the regimes discussed in the paper - the catapult mechanism, edge of stability, etc. - have already been demonstrated repeatedly in very similar models in multiple previous works (and I should give the credit to the authors for carefully listing them in two related work sections). The added value of the present paper - detailed analysis of the extended toy model and some predictions for realistic models - can certainly be of interest to a part of ML community, but just looks relatively marginal to me on the ICLR scale.

**Questions:**

Line 813: "constant width networks in SP with learning rate $\eta=\Theta(1/n)$ are equivalent to those in NTP Yang & Hu (2021)." - Where exactly is this shown in Yang & Hu (2021)? Table 1 and Figure 2 there actually show SP and NTP as different parameterizations.

---

> ### Author Response · Authors · 2024-11-19
>
> We thank the reviewer for their careful reading and detailed comments.
>
> > ... The UV model studied in the paper is a toy model and is only a slight generalization of the well-known model from Lewkowycz et al. (2020)
>
> While we use a similar model compared to Lewkowycz et al. (2020), our analysis differs in two key aspects:
>
> 1. Our work recognizes the crucial role of non-zero target $y$ and parameterization. Lewkowycz et al. (2020) set $y = 0$ (Eqs. 6-7 of their paper), which results in a model that cannot exhibit progressive sharpening (and consequently EoS) for any initialization or parameterization. In comparison, we observe progressive sharpening and EoS, precisely because of non-zero target $y$ and $\mu$P.
>
> 2. Furthermore, their analysis relies on specific approximations to simplify the dynamical equations to analyze catapult dynamics, particularly leveraging the existence of $\mathcal{O}(1/n)$ terms that can be ignored in the NTK regime. This approach cannot be generalized to $\mu$P ($n_{\text{eff}} = 1$) where such approximations are infeasible. In contrast, our generalized UV model dynamics comprehensively captures different sharpness phenomena across parameterizations.
>
> > All the regimes discussed in the paper - the catapult mechanism, edge of stability, etc. - have already been demonstrated repeatedly in very similar models.
>
> While we acknowledge that certain training regimes (such as catapult dynamics, progressive sharpening, EoS, etc.) have been demonstrated in prior works, our study is unique in providing a unified analysis encompassing all four regimes (especially the sharpness reduction regime) within a single model, which does not exist in the literature. Additionally, our work analyzes **practical conditions** under which different sharpness phenomena, such as early sharpness reduction, progressive sharpening, and EoS, are **not** observed.
>
>
> > Where exactly is this shown in Yang, Hu (2021)? Table 1 and Figure 2 there actually show SP and NTP as different parameterizations
>
> The $abc$ parameterization introduced by Yang & Hu (2021) exhibits a symmetry (Section 3.2). For any scalar $\theta \in \mathbb{R}$, the optimization trajectory remains invariant under the following simultaneous transformations: $a^l \rightarrow a^l + \theta$, $b^l \rightarrow b^l - \theta$, and $c^l \rightarrow c^l - 2\theta$. Starting from Standard Parameterization, if we consider $\theta = 1/2$, then from Table 1, we observe that NTP and SP are equivalent except for the first layer. This first layer difference arises because in this table, input dimension $d_{\text{in}}$ is considered to be $\mathcal{O}(1)$ wrt width. If we consider 'constant width' networks ($d_{\text{in}} = \mathcal{O}(n)$), then NTP and SP are equivalent parameterizations. We acknowledge that the phrase 'constant width' is confusing. Accordingly, we have clarified this in the updated version of the manuscript.
> The difference between NTP and SP in Fig. 2 of Yang & Hu (2021) perhaps arises from the difference in the first layer.

---

> > ### Author Response · Authors · 2024-11-25
> >
> > We thank the reviewer for their time in reviewing and constructive comments. We would greatly appreciate your thoughts on whether our responses have addressed your questions. We remain available to provide any additional clarifications if needed.

---

> > > ### Comment · Reviewer_Knqc · 2024-12-03
> > >
> > > I thank the authors for their reply. I agree with the provided arguments, so I'm raising my rating.

---

### Meta-Review · Area_Chair_o1Rd · 2024-12-05

**Metareview:**

This work investigates phenomena related to sharpness, emphasizing ones involving EoS, by studying a `minimal working' model of two-layer linear network. Concerns were raised regarding the scope and applicability of the results to realistic settings. The authors provided detailed responses and additional clarifications, addressing these and other key issues to the general satisfaction of the reviewers. The main strengths of the paper, as highlighted by the reviewers, include its solid theoretical analysis, valuable insights into fundamental optimization phenomena, and clear presentation. While some concerns about the broader applicability of the work to complex models remained, the reviewers generally agreed that the results are interesting already for the proposed model and align with the community’s interest in understanding sharpness and dynamics.

**Additional Comments On Reviewer Discussion:**

See above:

Concerns were raised regarding the scope and applicability of the results to realistic settings. The authors provided detailed responses and additional clarifications, addressing these and other key issues to the general satisfaction of the reviewers. The main strengths of the paper, as highlighted by the reviewers, include its solid theoretical analysis, valuable insights into fundamental optimization phenomena, and clear presentation. While some concerns about the broader applicability of the work to complex models remained, the reviewers generally agreed that the results are interesting already for the proposed model and align with the community’s interest in understanding sharpness and dynamics.

---

### Decision · Program_Chairs · 2025-01-22

Accept (Poster)